# Circular dichroism of quantum defects in carbon nanotubes created by photocatalytic oxygen functionalization

Finn L. Sebastian [1], Leon Kaminski [1], Christoph Bendel[2], Yohei Yomogida [3], Yuuya Hosokawa[4], Han Li [5,6], Sebastian Lindenthal[1], Benjamin S. Flavel [7], Kazuhiro Yanagi [4] & Jana Zaumseil [1] ✉

Control over the chiroptical properties of low-dimensional semiconductors is a promising route toward next-generation optoelectronics and photonics. With their helical chirality, single-wall carbon nanotubes (SWCNTs) offer a suitable framework for exploring chiral excitonic states. In addition to intrinsic, one-dimensional excitons, the targeted functionalization of SWCNTs with luminescent defects introduces zero-dimensional quantum states that enhance photoluminescence quantum yields and exhibit single-photon emission at room temperature. Here, we demonstrate that these defect states inherit the chirality of the respective SWCNT enantiomer, as evident from near-infrared circular dichroism. This observation is achieved by utilizing photocatalysis for efficient and versatile functionalization of SWCNTs with luminescent oxygen defects. The employed approach, based on anthraquinone derivatives as photocatalysts, is applicable to SWCNTs with different diameters, in aqueous or organic dispersions, with different surfactants, and even enables lateral patterning of defects in SWCNT networks. Low catalyst concentrations and the absence of cytotoxic metals or reactants make this functionalization method highly biocompatible. Introducing luminescent defects with uniform binding configurations in sorted nanotube enantiomers represents a key step toward chirality control of quantum defects in SWCNTs.

Structural chirality in low-dimensional semiconductors leads to the formation of chiral excitonic states, which are fundamental to chiroptical effects such as circular dichroism (CD) and circularly polarized luminescence (CPL)[1]. Chiral excitons have been studied extensively in two-dimensional systems such as transition metal dichalcogenides[2], their twisted bilayer structures[3], and hybrid organic-inorganic perovskites[4]. With the integration of chirality as an additional degree of freedom in nanoscale systems, the precise control of chiral excitons

and an understanding of the impact of their confinement or localization at lattice defects or edges have become essential. The question to what degree chirality persists in such confined states is particularly important for applications that rely on quasi zero-dimensional excitons, e.g., circularly polarized single-photon emission (SPE)[5].

Due to their helical chirality, single-wall carbon nanotubes (SWCNTs) represent a model system for exploring the properties of chiral excitons. Their structural diversity as well as their optical and

[1]Institute for Physical Chemistry, Universität Heidelberg, Heidelberg, Germany. [2]Institute for Inorganic Chemistry, Universität Heidelberg, Heidelberg, Germany. [3]Research Institute for Electronic Science, Hokkaido University, Sapporo, Hokkaido, Japan. [4]Department of Physics, Tokyo Metropolitan University, Hachioji, Tokyo, Japan. [5]Department of Mechanical and Materials Engineering, University of Turku, Turku, Finland. [6]Turku Collegium for Science, Medicine and Technology, University of Turku, Turku, Finland. [7]Institute of Nanotechnology, Karlsruhe Institute of Technology, Karlsruhe, Germany. ✉e-mail: zaumseil@uni-heidelberg.de

electronic properties continue to drive fundamental insights into the photophysics of nanoscale systems[6,7] and their application in optoelectronics[8,9], as near-infrared (NIR) single-photon emitters[10,11], for biological imaging[12–14], and optical sensing[15,16]. Highly optimized techniques for nanotube dispersion and sorting[17] are the prerequisite for all of the above. The separation of not only specific (n,m) species but SWCNT enantiomers currently presents the most advanced level of sorting[18,19]. SWCNT enantiomer separation has been achieved by density gradient centrifugation[20], wrapping with chiral polymers in organic solvents[21], aqueous two-phase extraction (ATPE)[19], and gel chromatography[22] where the enantiomeric purities can exceed 90%[23]. Enantiomer-separated SWCNTs were employed for chirality-selective recognition of amino acids[24], the investigation of their Raman optical activity[25], and enantiomer-specific DNA-wrapping kinetics[26]. As the interaction of circularly polarized light with helically chiral carbon nanotubes enables the observation of cross-polarized $E_{ij}$ optical transitions in CD spectroscopy, enantiomer-separated SWCNTs were also crucial for determining their complete exciton band structure[18].

Simultaneously with these advancements in enantiomer separation, the targeted covalent functionalization of SWCNTs with luminescent defects (also referred to as organic color centers or quantum defects) was discovered[27,28] and optimized for various applications[29]. These luminescent defects can be created through the controlled reaction of SWCNTs with reactive oxygen species[27,30,31] or aryl diazonium salts[28,32]. This covalent functionalization leads to changes in the local electronic band structure and facilitates exciton localization at the defect sites. New NIR photoluminescence (PL) peaks appear that are red-shifted in comparison to the intrinsic $E_{11}$ PL of SWCNTs and exhibit PL lifetimes on the order of hundreds of ps[33]. The energy difference between the intrinsic exciton and the defect emission (100 meV or more) corresponds to the optical trap depth ($\Delta E_{opt}$) of the defect. As free excitons are efficiently funneled to these luminescent defects, they also lead to higher PL quantum yields (PLQYs). The precise PLQY values and enhancement factors depend on the initial nanotube length, defect density (ideally 5 − 8 per μm), functionalization method, and dispersion solvent[28,31,32,34]. The combination of exceptional photostability and narrow-band emission in the second biological window render SWCNTs with luminescent defects as excellent nanoscale emitters for high-contrast deep-tissue imaging with unprecedented spatial resolution[13,35,36]. The nearly zero-dimensional state of luminescent defects and their deep trap depth enable NIR SPE from SWCNTs even at room temperature[37,38], making them promising candidates for quantum light sources at telecommunication wavelengths[10,11].

Despite the many potential applications of luminescent defects, the handedness of the functionalized SWCNTs and its impact on the chiroptical properties of the defects themselves has not yet been explored. As defect-trapped excitons remain localized on the nanotube lattice instead of the introduced functional group[39], the chiral nature of the SWCNT should be retained in the defect states. However, so far this has not been shown experimentally. Furthermore, the extent to which localized excitons inherit the chirality of the surrounding SWCNT is currently unknown. To answer this question, a highly efficient and tunable method for the introduction of quantum defects is necessary that proceeds under mild reaction conditions to prevent excessive side reactions. These demands can be fulfilled by photocatalytic functionalization approaches, which are still unexplored for SWCNTs.

Photocatalysis has been a rapidly expanding area within organic synthesis. Current developments are driven by its inherent compatibility with higher atom economy and sustainability of chemical transformations[40]. Furthermore, photocatalysis can be used to control reaction pathways and rates by adjusting the irradiation wavelength and intensity. Among the various organic photocatalysts, anthraquinones (AQs) are particularly suitable for oxidative processes and have been employed in alcohol and C-H bond oxidation as well as for light-driven hydrogen peroxide production[41,42]. AQs combine large absorptivity in the UV-Vis spectral region with high excited state redox potentials. They can be easily derivatized to tune their solubility in water or organic solvents, which, together with their low cost, makes them excellent candidates for scalable and versatile photocatalytic reactions[43].

Here, we introduce AQs as organic photocatalysts for the controlled and efficient functionalization of SWCNTs with luminescent oxygen defects. The metal-free functionalization reaction yields SWCNTs with narrow defect emission peaks, high PLQYs and can be applied to nanotubes with different diameters in aqueous or organic dispersions, at low and high concentrations, and even with spatial resolution for SWCNTs thin films. Using photocatalysis with AQ, the functionalization of concentrated dispersions of separated (6,5) and (11,−5) SWCNT enantiomers with high defect densities and without detrimental side reactions becomes possible and thus enables CD spectroscopy of these defects states. We find that the chiral properties of the nanotubes indeed persist within the luminescent defects and even enable us to determine the relative size of free and localized excitons in functionalized SWCNTs.

## Results
### Photocatalytic functionalization of SWCNTs

For the development of a photocatalytic functionalization strategy for SWCNTs, CoMoCAT raw material was subjected to aqueous two-phase extraction (ATPE) to prepare monochiral (6,5) SWCNTs (see Methods and Supplementary Fig. 1) stabilized by sodium dodecyl sulfate (SDS) in water[34]. Luminescent oxygen defects (with $E_{11}^*$ emission at 1120 nm) were introduced to the nanotube lattice at room temperature via a photocatalytic cycle with anthraquinone-2-sulfonate (AQS) that produced reactive oxygen species (ROS) as shown in Fig. 1a. UV irradiation (UVA, excitation wavelength $\lambda_{ex} = 365$ nm) of AQS leads to the formation of an excited-state anthraquinone (for a comparison of the AQS absorption bands and the spectral output of the illuminating light-emitting diode see Supplementary Fig. 2). In the presence of water and oxygen, the excited-state AQS transforms into a hydroquinone species under generation of ROS. While the precise mechanism of this step and the structure of reactive intermediates, especially in water are still a subject of debate, the generation of strong hydroxylation agents has been confirmed[44–46]. The created ROS directly attack the nanotube sidewalls to form luminescent oxygen defects as evident from the new defect emission peaks in the PL spectra[31,47]. Subsequent autoxidation of the hydroquinone under ambient conditions produces hydrogen peroxide and completes the catalytic cycle. To a very limited extent, continuous irradiation with UVA-light will also drive the photodissociation of the generated $H_2O_2$, which supplies additional reactive hydroxyl radicals[48]. The rate of ROS production under these mild conditions is low and thus enables a high degree of control over the SWCNT functionalization reaction.

The bright, red-shifted $E_{11}^*$ emission feature of the luminescent oxygen defects appears within minutes of UVA-irradiation of a dispersion of (6,5) SWCNTs in the presence of the photocatalyst AQS (concentration 6 μM). The absolute PL intensity of the narrow defect emission peak (full width at half maximum of 49 nm) increases with the duration of UV-light exposure and reaches over 3 times the value of the initial absolute $E_{11}$ peak intensity. Notably, no other emission features or broadening are observed in PL spectra or PL excitation (PLE) maps (see Fig. 1b), demonstrating exceptional selectivity of the photocatalytic functionalization with AQS towards a single oxygen defect binding configuration.

Continued UV-light irradiation of the (6,5) SWCNT dispersion with AQS further increases the $E_{11}^*$ defect emission until a maximum is reached followed by a decrease of the absolute PL intensity, while the $E_{11}^*/E_{11}$ PL intensity ratio continues to increase (see Supplementary

Fig. 3a, b). The reduction in overall PL results from the significant perturbation of the electronic structure of the SWCNTs at very high degrees of functionalization, which also impedes the formation and transport of mobile $E_{11}$ excitons to the luminescent defect sites. SWCNT samples collected after different irradiation times were used to determine their absolute PLQY (see Methods)[34,49] and confirmed that the photocatalytic functionalization method produces highly emissive (6,5) SWCNTs with PLQYs of up to 3.6% in aqueous dispersion at optimum defect density compared to 0.6% before functionalization (see Fig. 1c). The required duration of irradiation to reach the maximum PLQY can be extended by reducing the AQS concentration from 6 to 4 µM, thus offering more accurate control over the degree of functionalization. NIR PL images of pristine and functionalized SWCNT dispersions under identical conditions further visualize the almost six-fold enhancement of the PLQY (see Supplementary Fig. 3c). This

increase of SWCNT PLQY and thus brightness with functionalization is significantly higher than values obtained with other common methods for the introduction of oxygen defects[27,30,47]. It is similar to a recently reported metal-catalyzed functionalization reaction[31], but does not require removal of the catalyst before assessing the degree of functionalization.

**Functionalization mechanism and properties of oxygen defects**
To gain further insights into the mechanism of the photocatalytic functionalization, several reference experiments were performed. First, adding only AQS to the (6,5) SWCNT dispersion or prolonged exposure to UV-light (2 h) without the catalyst does not lead to any changes in the nanotube PL spectra (see Supplementary Fig. 4a). Irradiation with 525 nm light for 1 h at an AQS concentration of 6 µM also did not result in any functionalization, corroborating that it is

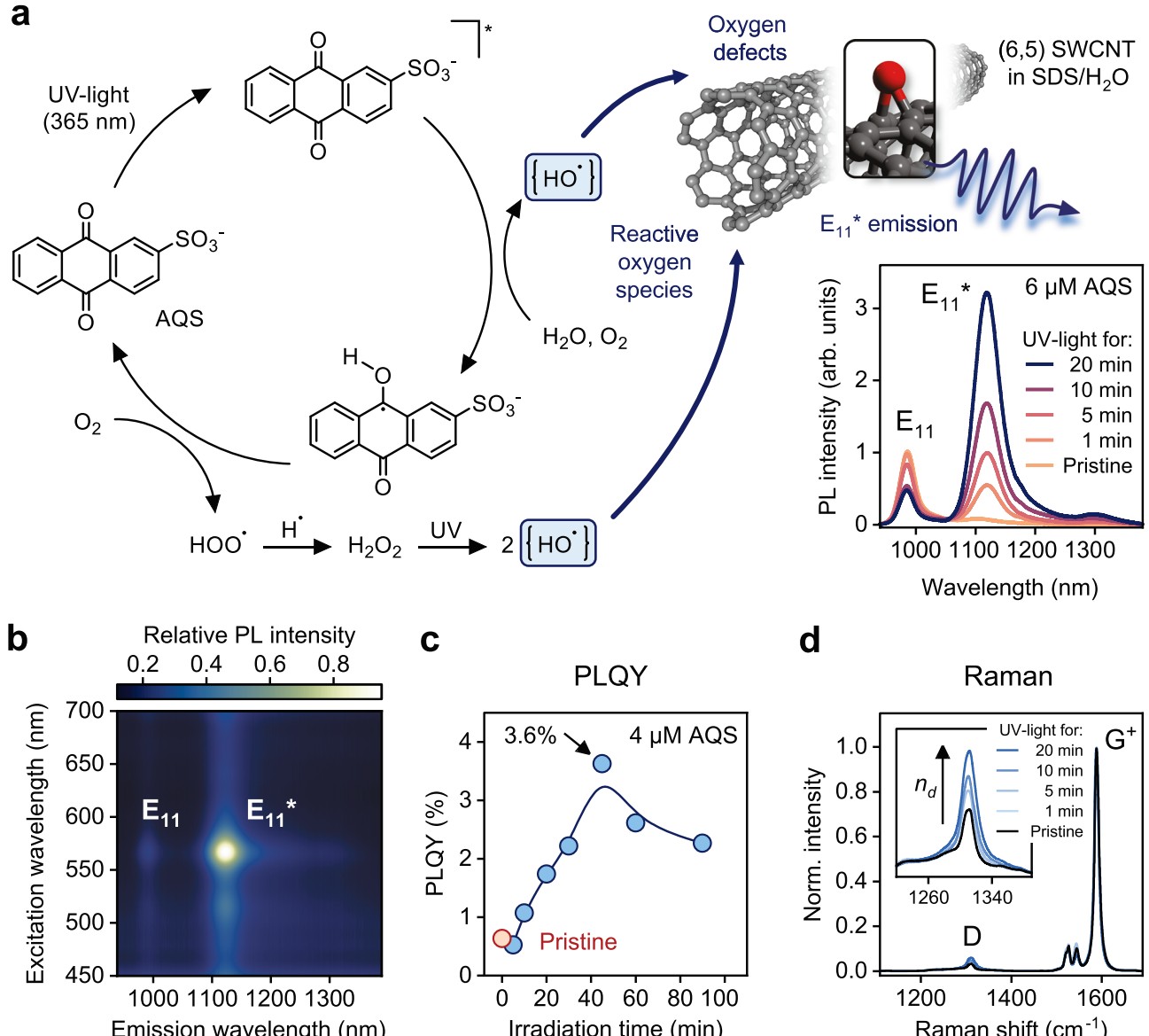

**Fig. 1 | Photocatalytic oxygen functionalization of (6,5) SWCNTs.**
**a** Photocatalytic generation of ROS using AQS (concentration 6 µM) as photocatalyst and UV-light ($\lambda_{ex}$ = 365 nm). Reaction of ROS with the SWCNT lattice creates luminescent oxygen defects leading to strong $E_{11}^*$ photoluminescence (PL, excitation at $E_{22}$ transition, $\lambda_{ex}$ = 570 nm), as fast-diffusing excitons are localized at the defect site. **b** PLE map of functionalized (6,5) SWCNTs (concentration AQS 6 µM, UV irradiation for 20 min) in aqueous dispersion (SDS, 0.33% w/v). **c** Evolution

of the total PL quantum yield (PLQY) of (6,5) SWCNT dispersions with increasing oxygen defect density (PLQY of the pristine sample: 0.6%, concentration AQS 4 µM, the blue line is a guide to the eye). **d** Averaged and normalized Raman spectra of (6,5) SWCNTs ($\lambda_{ex}$ = 532 nm) at different degrees of functionalization and detail of the Raman D-mode showing its relative increase in intensity compared to the $G^+$-mode with defect density $n_d$. Source data are provided as a Source Data file.

indeed the excitation of AQS that drives the reaction (see Supplementary Fig. 4b) and not the optical excitation of the nanotubes. A limited degree of functionalization through exposure to daylight for two days suggests that AQS remains catalytically active for an extended period of time (Supplementary Fig. 4c).

Stopping the UV-light exposure immediately interrupts the photocatalytic reaction, as visible from the identical PL spectra acquired directly after irradiation and 1 h later (see Supplementary Fig. 5a). Even after prolonged UV exposure of the dispersion, no further functionalization occurs after the irradiation is stopped, which suggests that short-lived reactive hydroxylation intermediates are responsible for the SWCNT functionalization and must be generated continuously. From a practical perspective, these short-lived ROS provide unparalleled reaction control, as continuous PL measurements are possible and no quenching or washing steps are required to shield the SWCNTs from reactants (e.g., by addition of sodium deoxycholate, DOC[31]). Supplementary Movie 1 shows the progressing photocatalytic SWCNT functionalization under UV irradiation and the simultaneous changes of the NIR PL of (6,5) nanotube dispersions with and without AQS. After reaching the desired PL brightness or $E_{11}/E_{11}^*$ ratio, the water-soluble catalyst and any potential reaction byproducts can be removed by spin-filtration to obtain pure nanotube dispersions (see Supplementary Fig. 5b).

The properties of luminescent oxygen defects created by photocatalytically produced ROS were further investigated by Raman spectroscopy. The resonant Raman spectra of functionalized (6,5) SWCNTs (see Fig. 1d) show an increase of the relative Raman $D/G^+$ mode ratio with increasing irradiation time. The Raman $D/G^+$ ratio is frequently used as a relative metric for defect densities in carbon nanotubes and recently has been expanded to determine absolute defect densities[34]. Importantly, Raman $D/G^+$ ratios for SWCNTs with luminescent oxygen and aryl defects scale differently in relation to the defect densities calculated from the decrease of the $E_{11}$ PLQY due to the formation of clusters in the case of oxygen defects[50]. This difference in scaling can help to distinguish between luminescent oxygen and aryl defects. For our photocatalytic approach, we find the same correlation between the Raman $D/G^+$ ratio and calculated defect densities as previously seen for oxygen defects. Maximum SWCNT PL intensities are obtained for ≈ 6 defect clusters per μm. While the $E_{11}^*$ emission wavelength (≈ 1120 nm, corresponding to an optical trap depth of 141 meV) already suggests the presence of oxygen defects[27,30,31,51], the $D/G^+$ scaling further corroborates the nature of the luminescent defects and excludes any direct reaction of the AQS with the nanotube lattice. This notion is further supported by examining the RBM/IFM region of the Raman spectra (see Supplementary Fig. 6)[50,52].

Luminescent quantum defects have significantly longer PL lifetimes compared to those of mobile $E_{11}$ excitons (few ps), as localization at the defect sites prevents fast non-radiative decay elsewhere on the nanotube lattice or at the nanotube ends[33,35]. Time-correlated single-photon counting (TCSPC) measurements of the $E_{11}^*$ PL decay at 1120 nm gave an amplitude-averaged lifetime of 158 ps (see Supplementary Fig. 7), which agrees well with literature data on luminescent defects[31,33,37]. Excitation density-dependent PL measurements further show the characteristic sub-linear scaling and eventual saturation of the $E_{11}^*$ emission (see Supplementary Fig. 8) due to defect state-filling[32]. Finally, temperature-dependent PL spectra yield a thermal trap depth $\Delta E_{\text{therm}}$ of 102 meV (see Supplementary Fig. 9), which is also in agreement with previous reports on oxygen defects[27,31,53]. The difference between the optical and thermal defect trap depth of ≈ 39 meV can be attributed to vibrational reorganization of the local nanotube lattice upon exciton trapping[53].

## Versatility of photocatalytic SWCNT functionalization

To explore the robustness and versatility of the presented photocatalytic functionalization of SWCNTs with AQS, different reaction conditions, nanotube environments and starting materials were tested. First, a range of common surfactants for the stabilization of (6,5) SWCNTs in water were employed leading to variations in the accessibility of the nanotube surface. Normalized PL spectra of (6,5) SWCNTs functionalized under otherwise identical conditions with SDS, DOC, sodium dodecylbenzene sulfonate (SDBS), or sodium cholate (SC) as surfactants at concentrations above their respective critical micelle concentration are shown in Supplementary Fig. 10. While no functionalization occurs in dispersions with DOC due to its dense surface coverage[54], comparable degrees of functionalization were achieved with the other surfactants. Dispersions with SDS showed the highest degree of functionalization in terms of the $E_{11}^*/E_{11}$ peak intensity ratio.

The small-diameter (6,5) SWCNTs (0.76 nm) are particularly well-suited for covalent functionalization, as the bond strain due to the lattice curvature leads to high reactivity. To investigate the effectiveness of the photocatalytic functionalization approach for other and especially larger diameter carbon nanotube species, (7,3), (8,3), (10,0), (9,2), (9,4), and (10,3) SWCNTs (diameter 0.71–0.92 nm) were sorted by gel chromatography[55] and transferred to SDS dispersions (see Supplementary Figs. 11 and 12 for absorption spectra and PLE maps). PL spectra of the different SWCNT species after functionalization in the presence of AQS and under UV-light irradiation are shown in Fig. 2a (see Supplementary Fig. 13 for corresponding PLE maps), confirming efficient functionalization even of larger diameter nanotubes without any changes to the reaction protocol. In agreement with data on luminescent aryl defects[56], an overall trend of decreasing optical trap depths for larger diameter nanotubes is observed (see Supplementary Fig. 14).

Their narrow-band PL in the second biological window (NIR-II) combined with high photostability have made SWCNTs promising fluorophores for biological imaging[12–14]. However, their application as emitters in biological environments or even living organisms requires a high degree of biocompatibility, which is usually achieved by biocompatible polymers such as ssDNA or pegylated phospholipids (PL-PEG)[24,36,57]. Since functional groups on the nanotube surface often disrupt surfactant coverage or polymer wrapping[31,38] and thus make post-functionalization surfactant exchange less efficient, protocols for functionalizing already biocompatible SWCNTs are highly desirable. As a proof of concept, pristine (6,5) SWCNTs were transferred by dialysis to PL-PEG$_{5000}$ and (GT)$_6$-ssDNA dispersions, respectively. Successful wrapping with these biocompatible polymers was corroborated by the significant redshifts of the corresponding $E_{11}$ emission and absorption peaks (see Supplementary Figs. 15 and 16). Without any additional treatment after dialysis, the PL-PEG-wrapped (6,5) SWCNTs were functionalized photocatalytically in the presence of AQS and emission from luminescent oxygen defects was evident. The integrated PL intensities were more than four times higher than pristine PL-PEG-wrapped nanotubes (see Fig. 2b). For (GT)$_6$-wrapped (6,5) SWCNTs, the photocatalytic functionalization was also possible but led to lower final $E_{11}^*/E_{11}$ PL intensity ratios (see Supplementary Fig. 16c). The apparently lower reactivity of ssDNA-wrapped SWCNTs may result from a difference in nanotube surface coverage[58]. Possible unwanted side reactions between ROS and the buffer system (PBS) may also contribute in addition to a lower colloidal stability due to alterations of DNA strands by ROS. The functionalization efficiency might be improved by optimizing the oligomer sequence or ssDNA concentration[59]. After functionalization, the AQS could be easily and completely removed from the dispersions by spin-filtration, ensuring biocompatibility of the final sample (see Supplementary Fig. 5b).

As the cost and effort to prepare monochiral SWCNT dispersions is still high, unsorted SWCNTs are often employed for biological imaging and sensing applications, as they can be obtained in high concentrations from commercially available raw materials (e.g., CoMoCAT)[12,60]. Oxygen-functionalized CoMoCAT SWCNTs have been shown to enable sensing of metabolites such as dopamine or

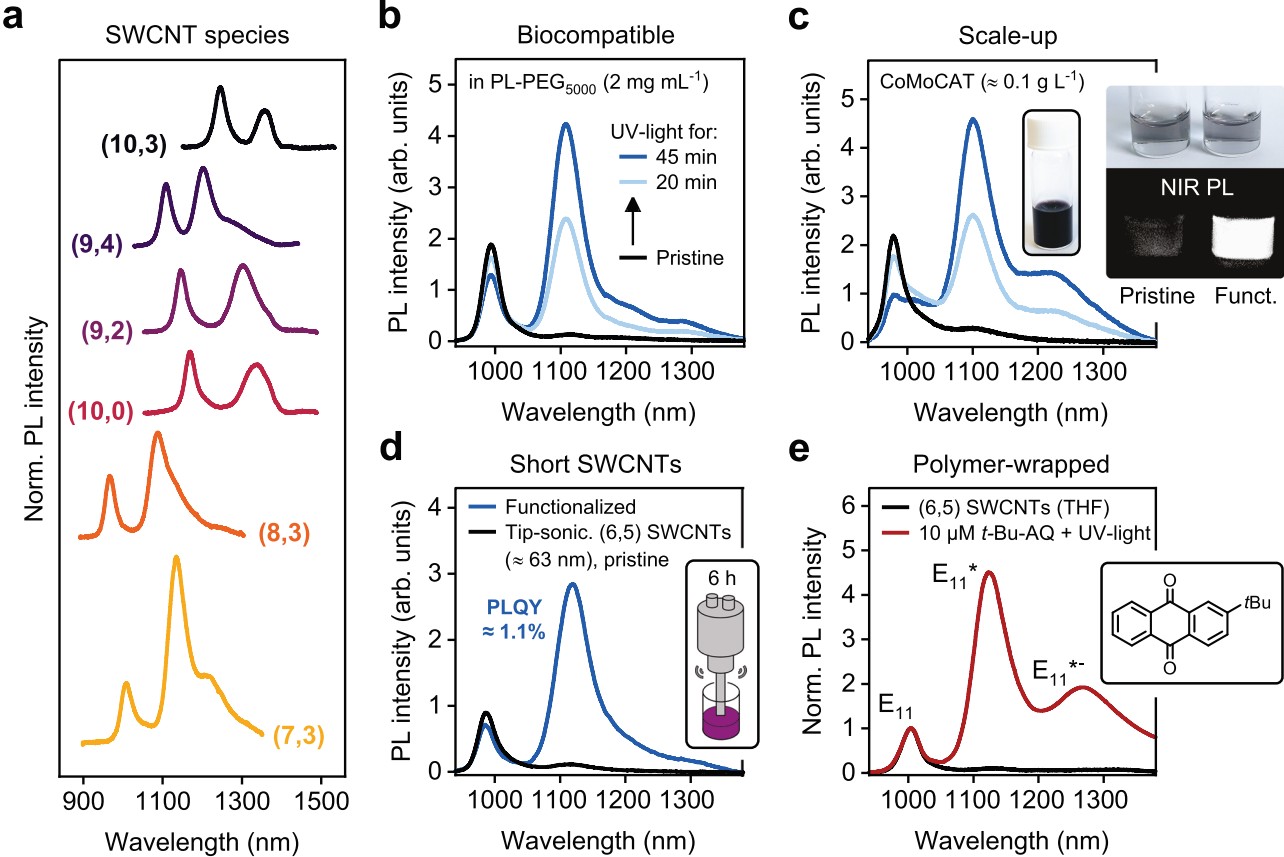

**Fig. 2 | Versatility of photocatalytic SWCNT functionalization. a** Normalized PL spectra of different oxygen-functionalized SWCNT species with diameters ranging from 0.71 nm to 0.92 nm (excitation at respective $E_{22}$ optical transition wavelengths as determined from UV-Vis-NIR absorption spectra, see Supplementary Fig. 11). **b** Functionalization of (6,5) SWCNTs after surfactant exchange to biocompatible polymer PL-PEG$_{5000}$ (concentration 2 mg mL$^{-1}$, without any post-treatment after dialysis). **c** Scale-up of photocatalytic functionalization using CoMoCAT raw material (SDS, 0.33% w/v) at high SWCNT concentrations ($\approx$ 0.1 g L$^{-1}$); black line: pristine SWCNTs; light blue: 10 min UV irradiation; dark blue: 20 min UV irradiation. The inset shows NIR PL images (950–1600 nm) of diluted pristine and functionalized CoMoCAT dispersions at identical concentration (525 nm excitation). **d** Functionalization of (6,5) SWCNTs shortened by tip sonication (6 h, average length of $\approx$ 63 nm). **e** Photocatalytic functionalization of PFO-BPy-wrapped (6,5) SWCNTs in THF using 2-*tert*-butylanthraquinone as photocatalyst. Source data are provided as a Source Data file.

cholesterol[60,61]. Their sensitivity and applicability could be enhanced by overall brighter luminescent defects. For this purpose, the photo-catalytic introduction of oxygen defects was scaled-up to highly concentrated CoMoCAT dispersions (0.1 g L$^{-1}$) in SDS (for absorption spectra and PLE maps see Supplementary Fig. 17). PL spectra of CoMoCAT SWCNTs before and after functionalization with AQS/UV-light illustrate a strong brightening effect (see Fig. 2c). The functionalized SWCNTs exhibit a 4.5× higher integrated PL intensity compared to the pristine CoMoCAT dispersion as also evident from the NIR PL images of pristine and oxygen-functionalized dispersions under the same conditions.

In addition to the choice of coating, short SWCNTs (<100 nm) are significantly more biocompatible than long nanotubes. They are also better suited for real-time imaging with high spatial resolution[62]. However, shortening nanotubes by oxidative cutting or ultrasonication severely reduces their PLQY due to quenching of mobile excitons at the nanotube ends[32,63]. Luminescent aryl and oxygen defects offer a way to alleviate this problem by efficient exciton localization[31,35]. To assess the brightening effect by photocatalytic functionalization for short SWCNTs, dispersions of pristine (6,5) SWCNTs (average length $\approx$ 580 nm) were subjected to 6 h of tip sonication, which shortened them to an average length of $\approx$ 63 nm (for absorption and PL spectra as well as AFM images, see Supplementary Figs. 18 and 19). As shown in Fig. 2d, irradiation with UV-light at an AQS concentration of 6 μM led to efficient functionalization with luminescent oxygen defects and an

exceptionally high PLQY of 1.1% (at optimal defect density) for such short nanotubes. With such brightness values these oxygen-functionalized SWCNTs are highly suitable for in-vivo, high-resolution imaging.

So far, all functionalization experiments were performed in aqueous dispersions. However, to create thin films, networks and finally devices with functionalized SWCNTs, the dispersions must be compatible with established film deposition techniques such as spin coating or aerosol-jet printing, which often rely on organic solvents[10,32,64,65]. The advantages of organic solvents compared to water in this context include their high vapor pressure, lower surface tension, and compatibility with processing under inert conditions. To test the photocatalytic functionalization of SWCNTs in organic solvents, 2-*tert*-butylanthraquinone (*t*-Bu-AQ) was chosen as photocatalyst due to its solubility in many organic solvents and its already established commercial application (e.g., in the industrial anthraquinone process)[42]. Monochiral dispersions of (6,5) SWCNTs were prepared by shear-force mixing (SFM) with a polyfluorene-bipyridine (PFO-BPy) copolymer in toluene, which yields long nanotubes with high PLQYs of 2% (for absorption spectra and PLE maps see Supplementary Figs. 20 and 21)[49]. When the prepared dispersions of (6,5) SWCNTs in toluene were mixed with *t*-Bu-AQ (10 μM) and exposed to UV-light, no functionalization reaction occurred (see Supplementary Fig. 22), presumably due to insufficient stabilization of the reactive hydroxylation species in a non-polar environment[66]. In contrast to that,

(6,5) SWCNTs that were dispersed in the more polar THF were easily functionalized and reached high $E_{11}*/E_{11}$ PL intensity ratios (see Fig. 2e).

The functionalization with oxygen defects instead of aryl $sp^3$ defects for polymer-wrapped SWCNTs in organic solvent was corroborated by the spectral position of the respective $E_{11}*$ peak at 1124 nm, with the expected blueshift by $\approx$ 40 nm compared to aryl defects[32]. In general, three possible binding configurations ($E_{11}*^+$, $E_{11}*$, $E_{11}*^-$) can be distinguished for luminescent oxygen defects based on previous theoretical studies[51]. The $E_{11}*^+$ ($\approx$ 1050 nm) and the more prevalent $E_{11}*$ ($\approx$ 1120 nm) PL peaks are associated with a longitudinally- and circumferentially-oriented ether-type binding motif (ether-L and ether-D, respectively). A further redshifted $E_{11}*^-$ PL peak ($\approx$ 1250 nm) results from an epoxide-like defect structure (epoxide-L). PL spectra of individual, oxygen-functionalized SWCNTs in a polymer matrix at cryogenic temperature (4.6 K, see Supplementary Fig. 23) indicated the presence of all three possible defect binding geometries, although the defect (binding configuration shown in Fig. 1a) with $E_{11}*$ emission is clearly dominant. Notably, a reduction of the selectivity towards the ether-D oxygen defect binding geometry in organic solvents compared to aqueous environments was evident from the additional, more redshifted $E_{11}*^-$ emission feature (epoxide-L defect configuration) in Fig. 2e. This constitutes the first report of intentional oxygen-functionalization of polymer-wrapped nanotubes directly in organic dispersions rather than on reactive oxide surfaces[37,67].

## Spatially resolved SWCNT thin-film functionalization

As the photocatalytic functionalization of SWCNTs with anthraquinones is driven by UV-light, it also allows for the spatially controlled introduction of oxygen quantum defects in thin films and networks of nanotubes. To date, the solid-state functionalization of SWCNTs with luminescent oxygen defects requires a direct and permanent contact between SWCNTs and reactive surfaces such as $SiO_2$ or metal oxides under inert conditions[37,67]. However this is often impractical for the fabrication of optoelectronic devices or SWCNT-coupled nanocavities, e.g., for collecting single photons from luminescent defects[8]. To perform photocatalytic functionalization of thin films of SWCNTs with AQS, we employed an irradiation setup as detailed in Fig. 3a. Glass or Si/$SiO_2$ substrates were coated with a thin and chemically inert layer of a crosslinked bisbenzocyclobutene polymer (BCB, $\approx$ 30 nm) to exclude any unintentional surface reactions or trap states[68]. Subsequently, polymer-wrapped (6,5) SWCNTs were spin-coated from toluene onto the BCB-covered substrates to achieve network densities similar to those for light-emitting field-effect transistors (see Fig. 3b)[9]. Finally, a shadow mask and a droplet of an aqueous AQS solution (2 mg mL$^{-1}$) were placed on top of the nanotube film. Subsequent irradiation with UV-light ($\lambda_{ex}$ = 365 nm) led to oxygen defect functionalization within minutes as visualized by Raman D/$G^+$ ratio mapping in Fig. 3c. Only areas that were not covered by the shadow mask were functionalized. The corresponding averaged PL spectra only show the intrinsic $E_{11}$ and narrow $E_{11}*$ defect PL (see Fig. 3d, e), confirming the high selectivity of the photocatalytic functionalization with AQS in water for creating a single oxygen defect binding configuration even in polymer-wrapped SWCNT networks on a solid substrate.

By applying an absolute defect quantification approach based on Raman spectroscopy[34,50], we calculated an average defect density of $\approx$ 13 $\pm$ 4 oxygen atom defects per µm of nanotube (integrated Raman $\Delta$(D/$G^+$) ratio of 0.03 $\pm$ 0.01). This value is within a reasonable range for the observed $E_{11}*/E_{11}$ PL intensity ratios of $\approx$ 1:1 (see Fig. 3e and Supplementary Fig. 24) and supports the assumption that no significant side reactions have occurred.

The spatial resolution of the functionalization method tested here was limited by the features of the shadow mask, as indicated by a nanotube thin film exposed with a stripe pattern (see Supplementary Fig. 24). However, the maximum spatial resolution of the process is most likely limited by the diffusion of the short-lived reactive hydroxylation species.

## Circular dichroism of luminescent oxygen defects

Chirality in low-dimensional semiconductors, such as hybrid organic-inorganic perovskites[1], quantum dots[69] or transition metal dichalcogenides[2,70] can be induced via geometrical distortion or supramolecular interactions with chiral ligands or additives. In contrast to that, SWCNTs exhibit intrinsic chirality across multiple length scales from few nm up to several µm as a direct consequence of the intrinsic helicity of the rolled-up carbon lattice. Except for zig-zag SWCNTs, all semiconducting SWCNTs are chiral and can be classified as left- or right-handed enantiomers according to their optical activity in CD spectra. The corresponding mirror image of a $(n,m)$ nanotube is a $(n + m, -m)$ nanotube, as exemplified by the SWCNT enantiomers (6,5) and (11,−5)[71].

The excitons in SWCNTs represent an intermediate regime between delocalized Wannier-Mott and tightly-bound Frenkel excitons due to their one-dimensional confinement. They exhibit coherence lengths of several nm but also large binding energies[72]. Consequently, SWCNTs offer a suitable framework for the characterization of free and defect-localized excitons and their interactions with a chiral carbon lattice. Due to limited light absorption by the few luminescent defects (5–30 defects per µm of nanotube) that are usually introduced by covalent functionalization[32] it was not yet possible to determine their chiroptical properties, e.g., by CD spectroscopy. The photocatalytic functionalization of SWCNTs enables the introduction of large numbers of oxygen defects. At the same time, the method proceeds with remarkably high selectivity for the ether-D ($E_{11}*$) binding configuration even in highly concentrated SWCNT dispersions. Thus, the $E_{11}*$ absorbance values and well-defined absorption peaks necessary to obtain reliable CD spectra become accessible.

To take advantage of this possibility, (6,5) and (11,-5) SWCNT enantiomers were separated by gel chromatography[18,22] and transferred to SDS dispersions (see Supplementary Figs. 25 and 26 for absorption spectra and PLE maps). CD spectra of pristine (6,5) and (11,−5) SWCNTs show the expected $E_{ii}$ and $E_{ij}$ transitions in the UV-Vis and NIR spectral range (see Fig. 4a, b). Enantiomeric purities of 84% for (6,5) and 89% for (11,−5) SWCNTs were determined from the normalized ellipticity ($CD_{norm} = CD_{E_{ii}} / Abs_{E_{ii}}$) at the $E_{22}$ transition[71].

For ideal signal-to-noise ratios in CD measurements, optical densities (OD) of $\approx$ 0.8 – 1 cm$^{-1}$ are desired. To obtain these values for the $E_{11}*$ transition, the pristine (6,5) and (11,−5) enantiomer dispersions were concentrated by spin-filtration to an OD of $\approx$ 3.5 cm$^{-1}$ at the $E_{11}$ transition. Subsequently, the SWCNT dispersions were functionalized with luminescent oxygen defects using AQS (10 µM) and UV irradiation. As this method enables frequent checks after each step of irradiation, direct tuning of the degree of functionalization was possible for both dispersions. The absorption spectra (see Fig. 4c) of both (6,5) and (11,-5) dispersions show the expected broadening of the $E_{11}$ transition upon extensive functionalization (see also Supplementary Fig. 27 for $E_{22}$ transitions) and an additional absorption peak at $\approx$ 1105 nm, which we attribute to direct absorption into the $E_{11}*$ defect state. The corresponding CD spectra (see Fig. 4d) also show a new CD feature with the same energy as the $E_{11}*$ defect absorption and the same sign as the $E_{11}$ transition. Hence, we can assign this new CD peak to the defect-related $E_{11}*$ exciton with clear evidence that the chirality of the nanotube also determines the chiroptical properties of the quantum defects.

As noted above, a large defect density is necessary to reach sufficient optical densities for CD measurements. The absolute quantification of the total defect densities of the functionalized (6,5) and (11,-5) SWCNT dispersions by Raman spectroscopy[50] yielded remarkably high values of about 70 oxygen atom defects per µm of SWCNT (see

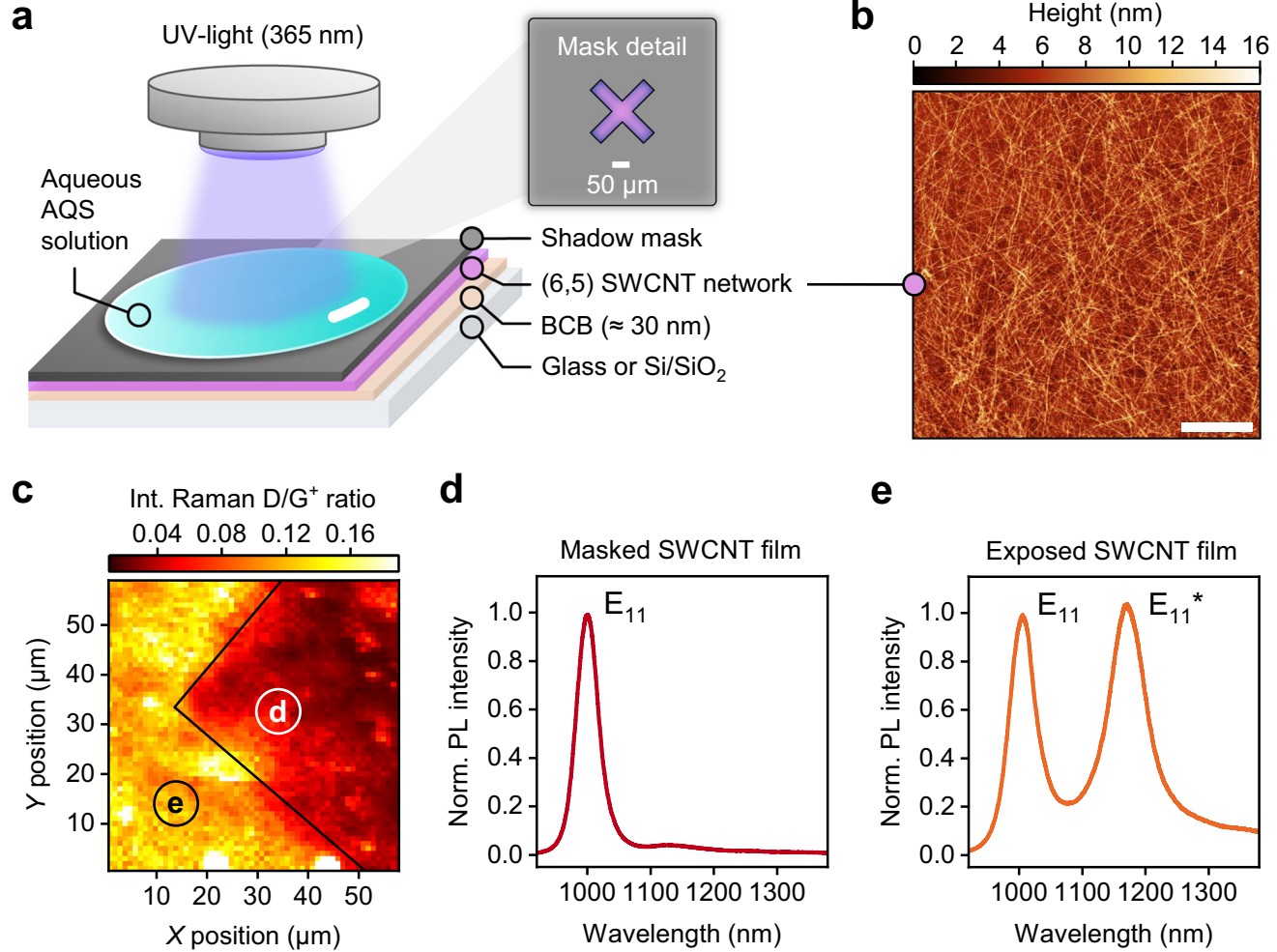

**Fig. 3 | Spatially resolved functionalization of SWCNT thin films. a** Schematic illustration of the functionalization procedure using UV-light ($\lambda_{ex} = 365$ nm) and a shadow mask, which is placed on top of a spin-coated nanotube network (polymer-wrapped (6,5) SWCNTs) on BCB-coated glass or silicon substrates. Irradiation through an aqueous AQS solution (2 mg mL$^{-1}$) in contact with the nanotube film leads to the formation of luminescent defects in the exposed areas. **b** AFM image of SWCNT network used for functionalization (scale bar, 1 μm). **c** Map of integrated Raman D/G$^+$ ratios at the shadow mask edge (solid black line) after functionalization. **d, e** Averaged, normalized PL spectra (>100 averaged spectra) of masked and exposed areas after functionalization. Source data are provided as a Source Data file.

Supplementary Fig. 28). Nevertheless, the corresponding PL spectra of these dispersions still show good spectral purity of the E$_{11}$* PL with only minor side band contributions (see Fig. 4c) while the E$_{11}$ emission is almost negligible and the E$_{11}$*/E$_{11}$ PL intensity ratio reaches more than 30.

Extracting the CD$_{norm}$ values (or absorption dissymmetry factors g$_{abs}$) of the E$_{11}$ and E$_{11}$* transitions from the CD and absorption spectra of the same SWCNT sample enables the direct quantification of the relative size of the free and localized exciton. Functionalized (6,5) and (11, −5) SWCNTs exhibit very similar CD$_{norm}$ values of −27 and 31 mdeg at the E$_{11}$ and -15 and 14 mdeg at the E$_{11}$* transition, respectively. From the approximately two-fold difference in CD$_{norm}$ values between the free and defect-localized exciton, a reduction of the exciton size by a factor of ≈ 2 upon trapping can be estimated (see Fig. 4e, and Supplementary Note 1). Such a contraction of the exciton upon confinement at luminescent quantum defects was previously proposed based on theoretical considerations[39,73]. The CD data presented here provides direct experimental evidence for this assumption and corroborates that the trapped E$_{11}$* exciton remains strongly coupled to the surrounding nanotube lattice. Thus, the broken mirror symmetry of the helically chiral SWCNT lattice is maintained even in the confined excitonic states of the quantum defects. Based on reported values of ≈

5–10 nm for the coherence length of the free exciton[74], our analysis suggests a confinement of the localized E$_{11}$* exciton to ≈ 2–5 nm, which is in good agreement with theoretical predictions[51]. Overall, the observation of CD in luminescent quantum defects provides a tool for exploring and exploiting the chiroptical properties of SWCNT enantiomers and could be utilized for chirality-sensitive optical readout schemes.

## Discussion
We have demonstrated a simple but highly efficient and versatile photocatalytic approach for the covalent functionalization of SWCNTs with luminescent oxygen defects. It is based on the photocatalytic activity of anthraquinones, which produces strong hydroxylation agents under UV illumination. These ROS react with the nanotube lattice to create oxygen defects of a specific binding type (ether-D), resulting in narrow defect emission peaks in the NIR. The reactivity of the system is controlled by variation of irradiation time and photocatalyst concentration, offering flexibility and scalability to create samples with precise defect densities, narrow defect emission, and high PL brightness (PLQY of 3.6%). No toxic, air-sensitive or metal-containing reagents are required for this simple photocatalytic transformation. Very low photocatalyst concentrations (μM) in combination

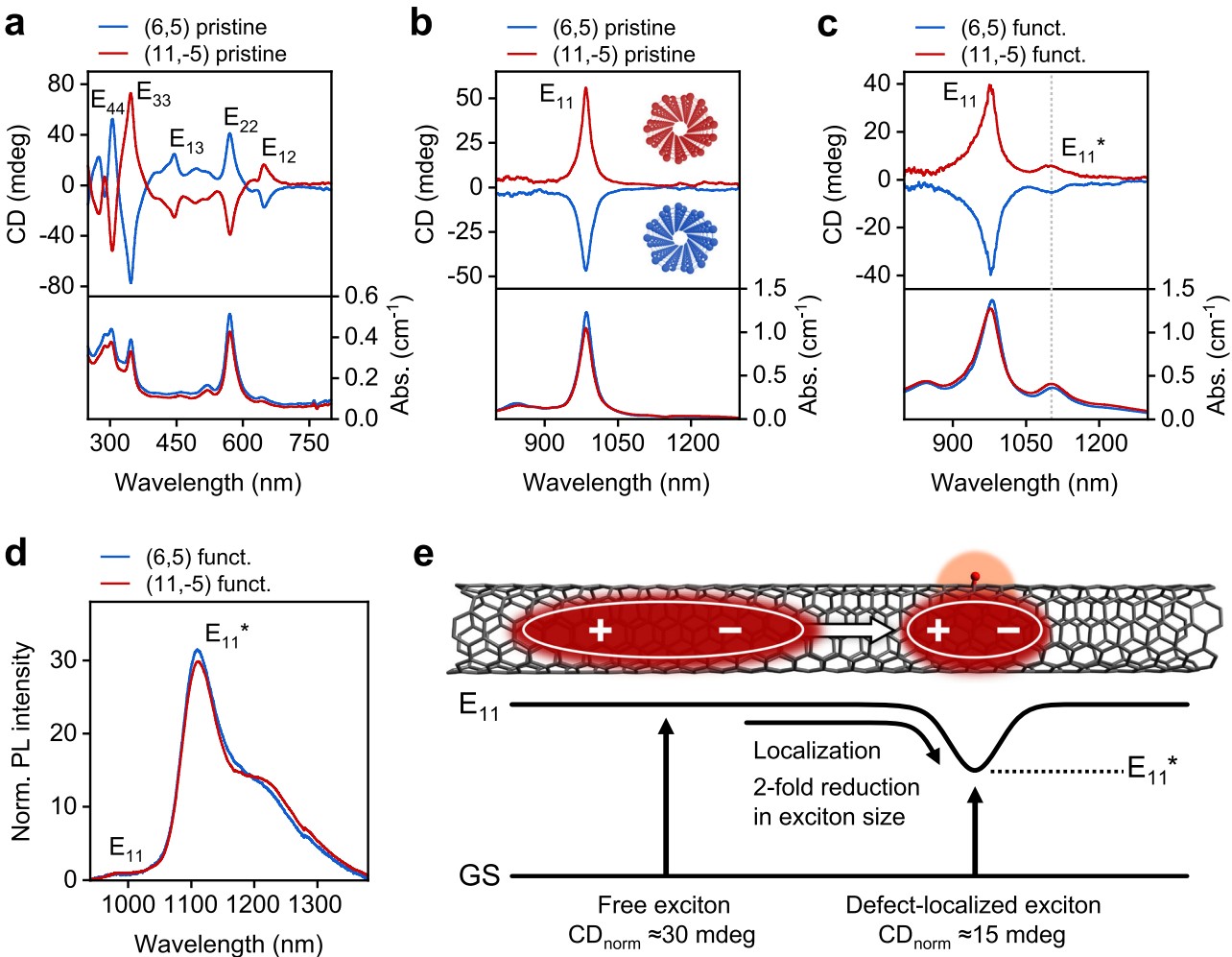

**Fig. 4 | Chirality of luminescent quantum defects. a** CD and absorbance spectra of pristine (6,5) and (11,−5) SWCNT enantiomers in the visible spectral region. **b** CD and absorbance spectra of pristine SWCNT enantiomers in the NIR spectral region (inset with front view of helically chiral SWCNT enantiomers). **c** CD and absorbance spectra of oxygen-functionalized SWCNT enantiomers in the NIR spectral region, demonstrating CD of $E_{11}^*$ defects. **d** PL spectra of oxygen-functionalized SWCNT dispersions in (**c**). **e** Schematic illustration of the reduction in exciton size by a factor of ≈ 2 upon confinement of the free exciton at the oxygen defect site as determined from the ratio of the respective $CD_{norm}$ values. Source data are provided as a Source Data file.

with their straightforward removal upon work-up are key advantages of this easy, sustainable and cost-efficient nanotube functionalization procedure.

The photocatalytic introduction of luminescent oxygen defects is suitable for various small-diameter SWCNTs (0.71 – 0.92 nm), providing access to photostable NIR emitters throughout the telecommunication O-band and NIR-II biological window. Our approach can be applied to SWCNTs in different surfactant systems, to biocompatible carbon nanotubes that are coated with pegylated phospholipids or DNA and to short (<100 nm) SWCNTs suitable for in-vivo bioimaging. It can be easily scaled-up for highly concentrated and unsorted SWCNT dispersions (e.g., CoMoCAT). Importantly, this broad versatility is achieved without any adaptations of the reaction protocol. Furthermore, a simple exchange of the water-soluble anthraquinone photocatalyst with a less polar derivative enables the functionalization of polymer-wrapped carbon nanotubes in THF. Even the spatially controlled introduction of luminescent oxygen defects in SWCNT networks is possible with high spectral purity and low levels of unfavorable side reactions, which is highly valuable for the fabrication of light-emitting devices, nanocavities, and integrated photonic circuits with NIR carbon nanotube emitters.

Finally, the photocatalytic introduction of oxygen defects in carbon nanotubes can be utilized to answer fundamental questions about the interactions of excitons in one-dimensional and chiral SWCNTs. The controlled functionalization of concentrated dispersions of (6,5) and (11,-5) SWCNT enantiomers with high defect densities allowed us to characterize the chiroptical properties of luminescent quantum defects in carbon nanotubes. The circular dichroism of the defect absorption with the same sign as the exciton absorption clearly demonstrates that the helical chirality of the nanotube lattice is retained in the quasi zero-dimensional defect state. This intrinsic chirality of the luminescent defects may enable circularly polarized light emission or chiral single photon emission from functionalized SWCNTs. Furthermore, the absorption dissymmetry factors calculated from the CD spectra give access to the relative sizes of the $E_{11}$ versus the $E_{11}^*$ excitons. The rather large mobile excitons in SWCNTs are reduced in size by a factor of two upon localization by the quantum defect. These values provide direct experimental insights into the fundamental properties of luminescent defects and can be used to further develop and benchmark theoretical models.

## Methods
### Preparation of (6,5) SWCNT dispersions
Monochiral (6,5) SWCNT dispersions in water were obtained by aqueous two-phase extraction (ATPE) from CoMoCAT raw material (CHASM SG65i-L58)[34]. For this purpose, a two-phase system of dextran

($M_W$ = 70 kg mol$^{-1}$, TCI) and polyethylene glycol (PEG, $M_W$ = 6 kg mol$^{-1}$, Alfa Aesar) was employed. The separation protocol used sodium deoxycholate (DOC, BioXtra, ≥ 98.0%) and sodium dodecyl sulfate (SDS, Sigma-Aldrich, ≥ 99.0%) for diameter-specific sorting of SWCNTs. Initially, the SDS concentration was increased to 1.1% (w/v) at a fixed DOC concentration of 0.04% (w/v) to transfer all species with diameters larger than (6,5) SWCNTs to the upper phase. Next, the SDS concentration was adjusted to 1.2% (w/v) and gradually increased to 1.5% (w/v) for the collection of fractions enriched in (6,5) SWCNTs. Metallic and semiconducting species were separated by addition of sodium cholate (SC, Sigma-Aldrich, ≥ 99.0%) and sodium hypochlorite as oxidant. Sorted (6,5) SWCNTs were concentrated in an ultrafiltration stirred cell (Millipore) using a 300 kg mol$^{-1}$ cut-off membrane, and adjusted to a 1% (w/v) SDS before further processing.

Highly-purified (6,5) SWCNT dispersions in toluene were obtained from CoMoCAT raw material (SG65i, Sigma-Aldrich, Lot No. MKC1004) via selective polymer-wrapping using shear-force mixing[49]. Briefly, the raw material was mixed with poly[(9,9-dioctyl-fluorenyl-2,7-diyl)-alt-(6,6'-(2,2'-bipyridine))] (PFO-BPy, American Dye Source, $M_W$ = 38 kg mol$^{-1}$, 0.5 g L$^{-1}$) in toluene, and dispersed by shear-force mixing (Silverson L2/Air, 10230 rpm, 72 h, 20 °C), followed by centrifugation (2×, 45 min, 60000 × $g$, Beckman Coulter Avanti J26XP) and filtration of the combined supernatants through a PTFE syringe filter (5 μm pore size).

### Sorting of additional SWCNT species
Monochiral (7,3), (8,3), (9,2), (9,4), (10,0), and (10,3) SWCNTs were obtained by gel chromatography employing a mixed surfactant system[55,75]. For (7,3), (10,0), (8,3), (9,2) SWCNTs, HiPco SWCNT raw material (NanoIntegris, Inc., diameter distribution 1.0 ± 0.3 nm) was dispersed in an aqueous solution of SC (0.8%, w/v) by ultrasonication (Branson Sonifier 250D) for 1–2 h. After ultracentrifugation at 210,000 × $g$ for 2 h, the supernatant was collected and mixed with aqueous solutions of SDS and SC to modify the surfactant concentration to 0.6% SDS + 0.4% SC. The separation was carried out with a HPLC system (Cytiva AKTAexplorer 10S) and columns (Cytiva HiScale 26/20) filled with gel beads (Cytiva Sephacryl S-200 HR) at a temperature of 17 °C. After loading with the SWCNT dispersion, adsorbed SWCNTs were eluted using a mixed surfactant solution of 0.6% SDS + 0.4% SC + $x$% sodium lithocholate (LC, >98%, Tokyo Chemical Industry Co., Ltd.). Adjustment of the LC concentration to 0.030%, 0.048%, 0.056%, and 0.065% yielded fractions highly enriched in (7,3), (10,0), (8,3), and (9,2) SWCNTs, respectively. For (9,4) and (10,3) SWCNTs, HiPco SWCNT raw material was dispersed in an aqueous solution of SC (1%, w/v) by ultrasonication for 2 h. After ultracentrifugation, the collected supernatant was mixed with aqueous solutions of SDS to modify the surfactant concentration (1.0% SDS + 0.5% SC). The separation was carried out with a HPLC system and columns (Cytiva HiScale 50/20) at a temperature of 20 °C. After loading with the SWCNT dispersion, adsorbed SWCNTs were eluted using a mixed surfactant solution of 1.0% SDS + 0.5% SC + $x$% DOC. The concentration $x$% of DOC was adjusted to 0.14% and 0.16% to obtain (9,4) and (10,3) SWCNT fractions, respectively. Prior to functionalization, the surfactant was exchanged to SDS (0.33%, w/v) by spin-filtration (Amicon Ultra-4, 100 kg mol$^{-1}$).

### Enantiomer separation of (6,5) and (11,-5) SWCNTs
Enantiomer separation of (6,5) SWCNT dispersions in water was performed by gel chromatography[22]. CoMoCAT SWCNTs (Sigma-Aldrich SG65, 704148) were dispersed in an aqueous solution of SC (1.0%, w/v) by ultrasonication (Branson Sonifier 250D). Following ultracentrifugation at 210,000 × $g$ for 2 h, the supernatant solution was collected and mixed with an aqueous solution of SDS to obtain a 2.0% (w/v) SDS + 0.5% (w/v) SC dispersion. The separation was performed in a two-step procedure, and the temperature was kept

constant at 20 °C. A high-performance liquid chromatography system (NGC, BIO-RAD) was equipped with columns filled with gel beads (GE healthcare Sephacryl S-200 HR) and the SWCNT dispersion was loaded into the first column equilibrated with 2.0% (w/v) SDS + 0.5% (w/v) SC. The non-adsorbed filtrate containing (6,5) and (11,-5) SWCNTs was collected and subjected to the second purification step. This fraction was combined with an aqueous solution of SC to prepare a 0.5% (w/v) SDS + 0.5% (w/v) SC dispersion, which was loaded into another column after equilibration with 0.5% (w/v) SDS + 0.5% (w/v) SC. Adsorbed SWCNTs were eluted using a mixed surfactant solution containing LC. A stepwise increase of the LC concentration yielded fractions containing 0.5% (w/v) SDS + 0.5% (w/v) SC + $x$% (w/v) LC, where high-purity (11,-5) and (6,5) SWCNTs were eluted at LC concentrations of 0.029% and 0.034%, respectively. Prior to functionalization, the obtained enantiopure dispersions were transferred to 0.33% (w/v) SDS by spin-filtration (Amicon Ultra-4, 100 kg mol$^{-1}$).

### Shortening of SWCNTs
Dispersions of pristine (6,5) SWCNTs (1% w/v SDS) were tip-sonicated for 6 h (Sonic Vibra Cell VCX500, amplitude 25%, on/off cycle 8 s/2 s) in a water-cooling bath (5 °C). After centrifugation at 60,000 × $g$ for 30 min (Beckman Coulter Avanti J-26S XP), the supernatant was removed and used for functionalization.

### Transfer of SWCNTs to PL-PEG$_{5000}$
Pristine (6,5) SWCNTs were transferred to PL-PEG$_{5000}$ using an adapted procedure as reported by Welsher et al.[36]. SWCNT dispersions were mixed with appropriate amounts of 18:0 PEG5000 PE (1,2-distearoyl-sn-glycero-3-phosphoethanolamine-N-[methoxy(polyethylene glycol)-5000], PL-PEG$_{5000}$, Avanti Polar Lipids, Inc.) to obtain a final concentration of 2 mg mL$^{-1}$. After dialysis against ultrapure water in a 1 kg mol$^{-1}$ dialysis bag (Spectra/Por, Spectrum Laboratories Inc.) for 5 days, the obtained dispersion was subjected to bath sonication for 10 min.

### Transfer of SWCNTs to ssDNA
For coating of pristine (6,5) SWCNTs with ssDNA, a protocol by Ackermann et al. was applied[57]. SWCNT dispersions (5 mL, OD at $E_{11}$ = 1 cm$^{-1}$) were mixed with 500 μL of (GT)$_6$ ssDNA (Sigma-Aldrich, 2 mg mL$^{-1}$) in phosphate buffered saline (1× PBS, Carl Roth) and dialyzed against PBS for 4 days (1 kg mol$^{-1}$ dialysis bag, Spectra/Por, Spectrum Laboratories Inc.). Following centrifugation at 20,000 × $g$ for 20 min (Hettich Micro 220 R), the supernatant was used without further treatment.

### Photocatalytic SWCNT functionalization in dispersion
Photocatalytic functionalization of aqueous SWCNT dispersions was carried out in standard borosilicate glassware. Typically, 4 mL screw cap vials were employed as reaction vessels. SWCNT dispersions were adjusted to a SDS concentration of 0.33% (w/v), followed by addition of appropriate amounts of a fresh stock solution of sodium anthraquinone-2-sulfonate (AQS, 1 mg mL$^{-1}$, Sigma-Aldrich, ≥ 98.0%) to achieve a final concentration of 6 μM. The reaction mixture was irradiated with UV-light (365 nm, 1.3 mW mm$^{-2}$, Thorlabs SOLIS-365C) until the desired defect density was reached. Spin-filtration (Amicon Ultra-4, 100 kg mol$^{-1}$) was employed for removal of AQS and transfer of functionalized SWCNTs to different surfactants (e.g., 1% (w/v) DOC for long-term storage).

The photocatalytic functionalization of PFO-BPy-wrapped (6,5) SWCNTs in THF was carried out similarly using 2-tert-butylanthraquinone (Sigma-Aldrich, ≥ 98.0%) at a final concentration of 10 μM in the reaction mixture. Before functionalization, the content of free PFO-BPy polymer in the dispersion was reduced by filtration of toluene dispersions of (6,5) SWCNTs through PTFE membranes (Millipore

JVWP, pore size 0.1 μm) and redispersion of the filter cake by bath sonication (20 min) in fresh THF.

## Photocatalytic functionalization of SWCNT thin films

Thin films of (6,5) SWCNTs were prepared by spin-coating ($3 \times 80$ μL, 2000 rpm, 30 s) of highly concentrated dispersions of PFO-BPy-wrapped (6,5) SWCNTs in toluene (OD at $E_{11} \approx 8$ cm$^{-1}$) onto BCB-passivated glass or Si/SiO$_2$ substrates[68], with washing (THF, isopropanol) and annealing steps (100 °C, 1 min) in-between.

Photocatalytic functionalization was performed by placement of a shadow mask above the thin film, followed by a droplet of an aqueous AQS solution (2 mg mL$^{-1}$). The desired functionalization density was controlled by varying the duration of UV-light irradiation (365 nm, 1.9 mW mm$^{-2}$, Thorlabs SOLIS-365C). Following removal of the shadow mask, the substrate was gently washed with ultrapure water and blow-dried with nitrogen.

## UV-Vis-NIR absorption spectroscopy

Baseline-corrected UV-Vis-NIR absorption spectra were acquired with a Cary 6000i UV-Vis-NIR absorption spectrometer (Varian Inc.). A background $S(\lambda) = S_0 e^{-b\lambda}$ was subtracted from absorption spectra to account for scattering[76,77].

## Raman spectroscopy

Raman spectra were recorded using a Renishaw inVia Reflex confocal Raman microscope in backscattering configuration equipped with a 50× long-working distance objective (Olympus, N.A. 0.5). For the determination of D/G$^+$ and IFM/RBM ratios, samples were prepared by drop casting of SWCNT dispersions onto glass slides (Schott AF32eco) and careful washing with ultrapure water for surfactant removal. Over 3000 individual spectra were recorded and averaged for each sample under 532 nm (D/G spectral region) or 785 nm (RBM region) laser excitation.

## Atomic force microscopy

Atomic force microscope (AFM) images were acquired with a Bruker Dimension Icon AFM under ambient conditions. For AFM images of SWCNT thin films, the instrument was operated in ScanAsyst mode, whereas AFM images for SWCNT length statistics were recorded in tapping mode. Samples for SWCNT length statistics were prepared by incubation of Si/SiO$_2$ substrates with aqueous poly-L-lysine hydrobromide (PLL, 0.1 g L$^{-1}$) for 10 min, followed by rinsing with ultrapure water. Diluted SWCNT dispersions (OD at $E_{11} = 0.1$ cm$^{-1}$) were placed on the PLL-covered Si/SiO$_2$ substrate for 10 min, followed by removal of the droplet and rinsing with ultrapure water.

## Photoluminescence spectroscopy (laser excitation)

For excitation power-dependent PL, PL lifetime, and PLQY measurements, a home-built setup was used as detailed in the following.

## Excitation-power dependent PL

For PL measurements at different excitation power, the wavelength-filtered output (570 nm) of a picosecond pulsed supercontinuum laser (NKT Photonics SuperK Extreme) was focused onto the sample with a 50× NIR-optimized objective (Olympus, N.A 0.65). Emitted light passed through a dichroic mirror (875 nm) and was guided through a long-pass filter (830 nm) to reject scattered laser light. An Acton SpectraPro SP2358 spectrograph (grating blaze 1200 nm, 150 lines mm$^{-1}$) coupled to a liquid nitrogen-cooled InGaAs line camera (Princeton Instruments, OMAV:1024) was used for spectra acquisition.

## PL lifetime measurements

Wavelength-resolved PL lifetime measurements were conducted in a time-correlated single-photon counting (TCSPC) scheme, and the output of the spectrograph was focused onto a gated InGaAs/InP avalanche photodiode (Micro Photon Devices) using a NIR-optimized 20× objective (Mitutoyo, NA 0.4). TCSPC histograms were obtained with a photon-counting module (PicoQuant PicoHarp 300) and biexponential reconvolution-based fits were used to fit the SWCNT $E_{11}$* PL decay (SymPhoTime 64 software). The instrument-limited $E_{11}$ excitonic decay in (6,5) SWCNT thin films was used to obtain the instrument response function.

## PL quantum yield determination

Spectrally-resolved PL quantum yields (PLQYs) of pristine and functionalized SWCNTs were obtained with an absolute method employing an integrating sphere[34,49]. The PLQY was determined as the ratio of emitted ($N_{em}$) to absorbed photons ($N_{abs}$):

$$\eta = \frac{N_{em}}{N_{abs}} \qquad (1)$$

A quartz glass cuvette (Hellma Analytics, QX) was filled with 1 mL of the analyte (solvent or SWCNT dispersion, OD at $E_{11} < 0.2$ cm$^{-1}$) and centered in an integrating sphere (LabSphere, Spectralon coating). Samples were excited at the respective SWCNT $E_{22}$ transition and the light exiting the integrating sphere was guided into the spectrometer with an optical fiber. Light absorption and scattering by the solvent were corrected for by repetition of the sample measurements using only the solvent. Integration of emission spectra yielded a value proportional to $N_{em}$. By subtracting the integrated attenuated laser signals of the sample and the pure solvent, values proportional to $N_{abs}$ were obtained. A stabilized tungsten halogen light source with known spectral power distribution (Thorlabs SLS201/M, 300 – 2600 nm) was employed for correction of the wavelength-dependent detection efficiency of the detector and absorption characteristics of optical components.

## Low-temperature single-nanotube PL spectroscopy

Low-temperature PL spectroscopy of individual SWCNTs was performed at 4.6 K in a closed-cycle liquid helium-cooled optical cryostat (Montana Instruments, Cryostation s50). Samples were prepared by dilution of oxygen-functionalized (6,5) SWCNT dispersions in THF to an optical density of 0.005 cm$^{-1}$ at the $E_{11}$ transition with a solution of polystyrene (PS, Polymer Source Inc., $M_W = 230$ kg mol$^{-1}$) to achieve a final PS concentration of 20 g L$^{-1}$. Spin coating (2000 rpm, 60 s) onto glass substrates coated with a thermally-evaporated layer of gold (150 nm) yielded individualized SWCNTs in a polymer matrix.

For acquisition of single-nanotube PL spectra, the output of a continuous wave laser diode (Coherent Inc., OBIS 640 nm, 1 mW) was focused onto the sample plane with a NIR-optimized 50× long-working distance objective (Mitutoyo, N.A. 0.42). Scattered laser light was rejected by placement of appropriate long-pass filters. The emitted light was guided to a grating spectrograph (Princeton Instruments, IsoPlane SCT-320, grating blaze 1200 nm, 85 grooves mm$^{-1}$). PL spectra were recorded with a thermoelectrically-cooled two-dimensional InGaAs camera array (Princeton Instruments, NIRvana 640ST). A custom script was employed for automated collection of PL spectra using a piezo-based nanopositioning system (Attocube ANC350, equipped with ANPx101 and ANPz102 nanopositioners).

## Photoluminescence spectroscopy and PLE mapping (lamp excitation)

Photoluminescence (PL) spectra and PL excitation-emission (PLE) maps were recorded with a Fluorolog-3 spectrometer (Horiba Jobin-Yvon) equipped with a liquid nitrogen-cooled InGaAs line camera (Symphony II) and a xenon arc-discharge light source (450 W). A Peltier-based temperature controller was used to perform

temperature-dependent PL measurements of liquid samples between 283 and 328 K.

## NIR PL imaging

PL images of SWCNT dispersions in the NIR (950–1600 nm) were recorded with a thermoelectrically-cooled InGaAs camera (Xenics XEVA-CL-TE3, 252 × 320 pixels) under excitation with LED light sources (Thorlabs SOLIS-365C or SOLIS-525C). Emitted light was guided through a long-pass filter (950 nm) and focused onto the 2D focal plane array using a tube lens.

## Circular dichroism spectroscopy

Baseline-corrected UV-Vis-NIR circular dichroism (CD) and corresponding absorption spectra of pristine and oxygen-functionalized dispersions of SWCNT enantiomers were recorded with a JASCO J-1700 CD spectrometer at a scanning speed of 50 nm min$^{-1}$.

## Reporting summary

Further information on research design is available in the Nature Portfolio Reporting Summary linked to this article.

## Data availability

The data that support the findings of this study are available from the heiDATA repository[78] and from the corresponding author upon request. Source data are provided with this paper.

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

## Acknowledgements

This project has received funding from the European Research Council (ERC) under the European Union's Horizon 2020 research and innovation programme (Grant Agreement No. 817494 "TRIFECTs", F.L.S., S.L., J.Z.).

B.S.F. acknowledges support by the Helmholtz Research Programs Natural Artificial and Cognitive Information Processing (NACIP) and Materials Systems Engineering (MSE). H.L. gratefully acknowledges support from the Turku Collegium for Science, Medicine and Technology (TCSMT, Starttiraha 26005020). Y.Y. acknowledges support from JSPS KAKENHI through Grant Nos. JP22H05468, JP22H01911, and JST FOREST No. JPMJFR235Z, and from Konica Minolta Imaging Science Foundation and Iketani Science and Technology Foundation, Japan. K.Y. acknowledges support from JSPS KAKENHI through Grant Nos. JP21H05017, JP23H00259, and JP24H01200, international joint research program (JPJSBP120252302) and U.S.-JAPAN PIRE through Grant No. JPJSJRP20221202, Japan. K.Y. and J.Z. acknowledge support by the ASPIRE-EXAR project through Grant No. JPMJAP2310, Japan. The authors acknowledge financial support by Heidelberg University for open access fees.

## Author contributions

F.L.S. fabricated samples, performed all measurements, and analyzed the data. L.K. and C.B. prepared selected samples and contributed to the development of the functionalization method. Y.Y. and Y.H. prepared enantiomer-separated (6,5) SWCNTs and other nanotube chirality dispersions under supervision of K.Y. H.L. and B.S.F. performed ATPE and provided (6,5) SWCNT dispersions in water. S.L. contributed to Raman measurements of SWCNT thin films. J.Z. conceived and supervised the project. F.L.S. and J.Z. wrote the manuscript with input from all authors.

## Funding

## Competing interests

The authors declare no competing interests.
