## [Transparent Peer Review file · Nature Communications]

Circular dichroism of quantum defects in carbon nanotubes created by photocatalytic oxygen functionalization

Corresponding Author: Professor Jana Zaumseil

Version 0:

Reviewer comments:

Reviewer #1

(Remarks to the Author)

The authors present a detailed experimental study on single-wall carbon nanotubes (SWCNTs) with luminescent defects (color centers) introduced via biocompatible photocatalytic chemical functionalization. The resulting defect emission exhibits clear chiral signatures, as evidenced by near-infrared circular dichroism. The insights gained from this study are valuable for achieving synthetic control over the optical chiral properties of nanotube materials. Given the significance of these findings, I recommend this work for publication in Nature Communications following a revision addressing the comments below. No further review is necessary.

1. RE introduction statement: "It is unclear whether the chirality of the extended excitonic system of the nanotube will persist in the confined, zero-dimensional state of a luminescent quantum defect." I believe this discussion requires further clarification. It is well established that the defect emission originates from the nanotube itself, rather than from an attached molecular group (here oxygen). More specifically, trapped excitons are localized within a 2–5 nm segment of the nanotube, where they undergo radiative decay. Given that this segment inherently possesses a chiral structure, it follows that chiral signatures in emission should be expected. However, what remains unclear is to what extent the chiral emissions from mobile excitons and trapped excitons are similar or different. The discussion should better reflect this distinction to provide a more precise framing of the research question... Well this discussion is present later... but it would be nice to give a flavor of it early.

2. Fig. 1a. I wonder if one could relate different PL curves to irradiation time to make better connection to PLQY points in Fig. 1c. Perhaps similar colors can be used for these curves and points.

3. The photocatalytic oxygen functionalization presented in this study is an important novelty, distinguishing it from well-established synthetic methods, such as the original Wiesman approach using ozonation. For SWCNTs produced via ozone/light fabrication, several studies have linked defect emission to the specific structure of oxygen binding sites, highlighting the importance of defect geometry (eg ref. 50). Along these lines, I wonder if the authors could expand their discussion on the structural characteristics of the binding sites introduced through their photocatalytic functionalization process (in addition to the discussion of similarities of optical characteristics), which perhaps should go beyond a simple statement "indicated the presence of all three possible defect binding geometries".

4. The above discussion will be important since there is a point of confusion: the next section states "Since the photocatalytic functionalization of SWNTs allows us to introduce large numbers of oxygen defects with a single binding configuration even in highly concentrated SWCNT dispersions." Do you mean that, out of 3, only a single configuration is prevalently luminescent?

5. Discussion of chiral properties and measurement of a confined exciton size is nice, indeed! Here I would suggest to introduce a couple of clarifications

(i) Definition of 'exciton size'. Some people define this quantity as a spatial distribution of center of mass of exciton (i.e. exciton wavefunction delocalization on the tube). Others think about average distance between photoexcited electron and hole (aka 1-2nm). Both are correct since placement of 2 particles needs to be characterized by 2 numbers. I believe you apply the former definition.

(ii) Another point that perhaps needs to be clarified is logic behind “CDnorm values for the free and defect localized exciton indicates a contraction of the exciton size by a factor of ~2”, which is rationalized by “the resulting increase in oscillator strength” and finally “the trapped E11* exciton has a reduced, but non-vanishing interaction with the chiral carbon nanotube lattice” To me (as evident from any quantum-chemical calculations), the oscillator strength of E11 exciton is about 2 times larger than that of E11* (i.e. the oscillator strength is proportional to the exciton size). This is barring any momentum considerations and applies only to ‘still’ excitons. I have problems with applying Miyauchi et al. arguments since they are operating with E11 (X11) size of 2 nm and E11* (X11*) size of 0.8 nm. This assumes true quantum confinement (electron and hole are squeezed like in quantum dot) and true 0-dimension system. Altogether this makes appearance of a giant oscillator strength, i.e. an increase of this quantity. Your sizes are different (5-10 vs 2-5 nm), and the electron and hole are not quantum confined. I personally would interpret your results as decrease of CDnorm values by a factor of 2 assumes reduction of the oscillator strength and subsequently a reduced exciton size by factor of 2, while the trapped E11* exciton has a similar to E11 interaction with the chiral carbon nanotube lattice. Again I would look at the orbital picture of E11 and E11* - they have the same wavefunction structure mimicking the tube chirality. I do not think that one needs to be definitive at this point and rather postulate a challenge to subsequent theoretical studies that will follow your work and help rationalize findings.

Reviewer #2

(Remarks to the Author)

The manuscript by Sebastien et al. demonstrated a novel photocatalytic functionalization method for generating oxygen defects in surfactant functionalized SWCNTs using ROS, anthraquinones, which react with the nanotube lattice under mild conditions. The level of reactivity of the ROS can be controlled easily, offering flexibility and scalability. They show that the approach is suitable for (6,5) purified and non-purified SWCNTs, for other different purified chiralities, enantiomers and even in organic solvents. The paper is well written and demonstrates several methods as a control to prove that oxygen defects occur. Moreover, the approach can be scaled to high SWCNT concentrations and allows spatially resolved functionalization in thin-film SWCNT networks, with potential applications in quantum emitters, bioimaging, and light-emitting devices. Overall, the work presents a scalable, tunable, and biocompatible photocatalytic approach for introducing luminescent oxygen defects in SWCNTs, demonstrating that these defects retain the chiroptical properties of the nanotube. Still, some issues need to be addressed before publication, as detailed below:

- Details about the excitation/irradiation parameters are missing in the results shown in Fig. 1a (excitation wavelength) and Fig. 1b (duration of UV irradiation).
- The authors mention six-fold enhanced emission of functionalized SWCNT dispersions at identical concentrations. Is this a general result? Which condition was tested? The details are missing in the text. How was this enhancement quantified?
- In Fig. 1d, a legend with the corresponding irradiation time is missing.
- What is PSB in Fig. S1a?
- In Fig. 2a, the excitation wavelength used for each chirality is missing. Moreover, it would be beneficial to show excitation-emission maps of these separated samples before and after functionalization in the SI.
- The authors mention no reaction occurred in toluene. Can the author include the SWCNTs emission spectra before and after UV-irradiation?
- Additionally, why did the authors use SWCNTs in toluene for the thin-film study, rather than THF?
- The authors exchange the functionalization of SWCNTs through dialysis to PEG and DNA. They should include the before- and after-absorption spectra.
- The authors mention “After functionalization, the AQS could be easily and completely removed from the dispersions, ensuring biocompatibility of the final sample.” How was this achieved and verified?

Minor points:

- “The corresponding mirror image of a (n,m) nanotube is a (n+m,-m) nanotube, as illustrated for the enantiomers (6,5) and (11,-5)”. I am assuming the authors intended to refer the reader to a particular figure?
- Typo: SWNT (on page 18).
- Typo: investigated It (page 4).
- Typo: R2 (subscript, page 12 in SI)
- In the SI, you refer to the panels as sometimes as a, b, c, and sometimes as (a), (b), (c) in the captions.

Reviewer #3

(Remarks to the Author)

This manuscript presents a well-executed and comprehensive study on photocatalytic functionalization of SWCNTs with luminescent oxygen defects, demonstrating for the first time D associated with these defect states. The methodology is versatile, biocompatible, and broadly applicable across various SWCNT chiralities and environments. The observation that E11* transitions retain the chiroptical properties of the host nanotube is significant and opens new directions for chiral optoelectronics and quantum emitters. However, the manuscript requires revision to clarify several important points before it can be considered for publication.

First, the authors should address whether circular dichroism in PL was observed or attempted. While the CD of defect-state absorption is compelling, circular dichroism in PL would provide further evidence of chirality retention in the emissive state and is highly relevant for future applications. If not observed, a brief discussion of experimental limits or constraints would be

valuable.

Second, the authors should explicitly discuss the role of symmetry breaking in enabling CD from localized states. Since chirality in CD arises from broken mirror symmetry, the defect-localized exciton must maintain coupling to the chiral lattice.

Third, the interpretation of reduced CDnorm as a measure of exciton size should be supported by reference to a quantitative or semi-quantitative model. A basic discussion of rotational strength or dipole coupling theory would lend physical credibility to this conclusion.

Lastly, the authors should consider the possibility that conformational flexibility of functional groups (e.g., rotation around single bonds) may reduce the observed CD. If different functional groups were tested or compared, any trends in CD response should be mentioned. Otherwise, this limitation should be acknowledged as a factor in interpreting the results.

In summary, the study is of high quality and interest, but requires clarification and modest theoretical framing to fully support its conclusions. I recommend major revision.

Reviewer #4

(Remarks to the Author)
Comments to the Author(s)

1. In Fig. 1, only in Fig. 1c the concentration of ACQ is given, but in the other figures it is unclear whether it is 6 μM as stated in the text. As the concentration of ACQ in the photocatalyst is a particularly important element of this paper, it is necessary to clearly indicate at which concentration the data in each figure was measured.

2. In page 8, Although the E11*/E11 PL intensity ratio is described as increasing in Fig S3, it is not clear whether the E11*/E11 PL intensity ratio was increasing as there were only separate data for E11* PL intensity and E11 PL intensity. Please add information on the E11*/E11 PL intensity ratio.

3. As shown in Figure S7, an increase in oxygen defects in SWCNTs should increase the fluorescence lifetime, so the value of the E11*/E11 PL intensity ratio, an indicator of oxygen defects, should be given. In addition, the relationship between the fluorescence lifetime, τ and the E11*/E11 PL intensity ratio should be shown. Does a higher E11*/E11 PL intensity ratio, for example as shown in Fig 1a, result in a longer fluorescence lifetime?

4. In page 12, the surface coverage of SWCNTs by PL-PEG or ssDNA is determined solely by the red-shift in the fluorescence spectrum, but additional information on the uniformity and thickness of those coatings should be added by morphological analysis, such as TEM or SEM. This is because the uniformity and thickness of the coating is expected to affect optical properties such as E11*/E11 PL intensity ratio.

5. In addition to the above, it is essential to explain the clear reason for the difference in E11*/E11 PL intensity ratio between PEG-coated SWCNTs and DNA-coated SWCNTs.

6. In Fig. 2c, a legend similar to Fig. 2b should be written to explain each fluorescence spectrum and an explanation should be added in the text.

7. In Figure S17, AFM images after sonication as well as AFM images before sonication should be appended for comparison. In addition, zoomed-in images of about 1 μm \times 1 μm are also needed to show fragmented SWCNTs of several tens of nm.

8. The E11* signal of the pre-functionalized (6,5) SWCNTs of the PLM maps shown in Fig S23 is already stronger than for the enantiomeric (11,-5) SWCNTs, what is the origin of this difference? Also, although CD and PL of the enantiomeric functionalized SWCNTs in E11* transitions shown in Fig 4c and d have been compared, the reviewer considers that the correct assessment may not be possible due to enantiomeric differences in the E11* signal in the pre-functionalised SWCNTs.

9. In page 10, the fluorescence lifetime values in SI Fig 7 (158 ps) differ from the values (153 ps) in the text. Please correct either to the correct value.

In page 20, the figure 4c is mistakenly written as Figure 4d.

Version 1:

Reviewer comments:

Reviewer #1

(Remarks to the Author)

The authors have adequately addressed all of the reviewers' comments, and I believe the manuscript is now suitable for publication in Nature Communications in its current form.

Reviewer #2

(Remarks to the Author)

The authors have addressed all my comments, and I am happy to recommend publication.

Reviewer #3

(Remarks to the Author)

As Reviewer #3, I have carefully reviewed the authors' rebuttal and revised manuscript. I am pleased to report that the authors have satisfactorily addressed all my concerns through their detailed responses and manuscript revisions. The additional analyses and clarifications provided have strengthened the manuscript's scientific rigor and clarity. Based on these improvements, I strongly recommend publication of this work in Nature Communications. The manuscript now presents a clear and compelling contribution to the field.

Reviewer #4

(Remarks to the Author)

Point-by-Point Response

Manuscript # NCOMMS-25-07588

We thank all reviewers for their careful consideration of our manuscript. We addressed all questions and remarks with additional data, figures, discussions, and references as detailed in the following point-by-point response. Any changes that were made to the revised manuscript or supplementary information are highlighted in **bold**. The revised versions of the manuscript and supplementary information are provided separately for review.

REVIEWER COMMENTS

Reviewer #1 (Remarks to the Author):

The authors present a detailed experimental study on single-wall carbon nanotubes (SWCNTs) with luminescent defects (color centers) introduced via biocompatible photocatalytic chemical functionalization. The resulting defect emission exhibits clear chiral signatures, as evidenced by near-infrared circular dichroism. The insights gained from this study are valuable for achieving synthetic control over the optical chiral properties of nanotube materials. Given the significance of these findings, I recommend this work for publication in Nature Communications following a revision addressing the comments below. No further review is necessary.

1. RE introduction statement: “It is unclear whether the chirality of the extended excitonic system of the nanotube will persist in the confined, zero-dimensional state of a luminescent quantum defect.” I believe this discussion requires further clarification. It is well established that the defect emission originates from the nanotube itself, rather than from an attached molecular group (here oxygen). More specifically, trapped excitons are localized within a 2–5 nm segment of the nanotube, where they undergo radiative decay. Given that this segment inherently possesses a chiral structure, it follows that chiral signatures in emission should be expected. However, what remains unclear is to what extent the chiral emissions from mobile excitons and trapped excitons are similar or different. The discussion should better reflect this distinction to provide a more precise framing of the research question... Well this discussion is present later... but it would be nice to give a flavor of it early.

RESPONSE:

We agree with the reviewer, a chiral component of the defect state should indeed be expected based on the fact that trapped excitons are still localized on the chiral nanotube lattice. Our work aims to experimentally confirm this chiral property of the defect state in absorption (not emission) and investigate to which degree differences between free and confined excitons on helically chiral nanotubes exist.

We have rephrased the corresponding introductory statement to highlight that while a chiral component of the E_{11}^* exciton is expected, its presence and extent have not yet been demonstrated and quantified experimentally (pages 4/5 of the revised manuscript).

“... As defect-trapped excitons remain localized on the nanotube lattice instead of the introduced functional group³⁹, the chiral nature of the SWCNT should be retained in the defect states. However, so far this has not been shown experimentally. Furthermore, the extent to which localized excitons inherit the chirality of the surrounding SWCNT is currently unknown.”

2. Fig. 1a. I wonder if one could relate different PL curves to irradiation time to make better connection to PLQY points in Fig. 1c. Perhaps similar colors can be used for these curves and points.

RESPONSE:

The PL curves in Fig. 1a and the PLQY presented in Fig. 1c are data from different batches of functionalized SWCNTs. For the experimental series of SWCNT functionalization shown in Fig. 1a, a catalyst concentration of 6 μM was used to reach maximum absolute PL intensities after ~ 20 min of irradiation time, whereas the catalyst concentration for obtaining PLQY samples was deliberately reduced to 4 μM to achieve more precise control over the degree of functionalization.

We have clarified this difference in the Figure caption. The used catalyst concentrations were added in Figs. 1a and 1c. We also added a legend, which indicates the duration of irradiation in Fig. 1a.

3. The photocatalytic oxygen functionalization presented in this study is an important novelty, distinguishing it from well-established synthetic methods, such as the original Wiesman approach using ozonation. For SWCNTs produced via ozone/light fabrication, several studies have linked defect emission to the specific structure of oxygen binding sites, highlighting the importance of defect geometry (eg ref. 50). Along these lines, I wonder if the authors could expand their discussion on the structural characteristics of the binding sites introduced through their photocatalytic functionalization process (in addition to the discussion of similarities of optical characteristics), which perhaps should go beyond a simple statement “indicated the presence of all three possible defect binding geometries”.

RESPONSE:

Following the reasoning in the mentioned reference (*ACS Nano* **2014**, 8, 10782), we assigned an **ether-D** binding configuration to the luminescent oxygen defects created by the photocatalytic functionalization process (E_{11}^*). The additional presence of E_{11}^{*+} (ether-L), E_{11}^* (ether-D), and E_{11}^{*-} (epoxide-L) spectral signatures in PL spectroscopy of single SWCNTs at cryogenic temperatures is another strong indication of oxygen defects instead of other sp^3 defects.

We have outlined this assignment in more detail and expanded our discussion on the binding configuration of the introduced luminescent oxygen defects in the respective section of the manuscript (see page 15 of the revised manuscript).

“... In general, three possible binding configurations (E_{11}^{+} , E_{11}^* , E_{11}^{*-}) can be distinguished for luminescent oxygen defects based on previous theoretical studies.⁵¹ The E_{11}^{*+} (~1050 nm) and the more prevalent E_{11}^* (~1120 nm) PL peaks are associated with a longitudinally- and circumferentially-oriented ether-type binding motif (ether-L and ether-D, respectively). A further redshifted E_{11}^{*-} PL peak (~1250 nm) results from an epoxide-like defect structure (epoxide-L).”*

4. The above discussion will be important since there is a point of confusion: the next section states “Since the photocatalytic functionalization of SWNTs allows us to introduce large numbers of oxygen defects with a single binding configuration even in highly concentrated SWCNT dispersions.” Do you mean that, out of 3, only a single configuration is prevalently luminescent?

RESPONSE:

All three oxygen defect binding configurations are luminescent (see *ACS Nano* **2014**, *8*, 10782, *Nat. Commun.* **2019**, *10*, 2874, *ACS Nano* **2024**, *18*, 20667, *Nanoscale Horiz.* **2024**, *9*, 2286), but a higher selectivity of the applied functionalization approach towards one of the three binding configurations will lead to PL spectra with mainly one PL peak corresponding to that configuration. In our case, we observe a high selectivity for the E_{11}^* defect binding geometry (ether-D binding configuration), even for highly concentrated samples and at high defect densities.

As this selectivity is a prerequisite for obtaining high-quality CD spectra, we have rephrased the respective statement to highlight that the combination of high selectivity for one binding configuration and the capability to introduce large numbers of defects are crucial to finally obtain CD spectra of the E_{11}^* optical transition (see page 19 of the revised manuscript).

“... The photocatalytic functionalization of SWCNTs enables the introduction of large numbers of oxygen defects. At the same time, the method proceeds with remarkably high selectivity for the ether-D (E_{11}^) binding configuration even in highly concentrated SWCNT dispersions. Thus, the E_{11}^* absorbance values and well-defined absorption peaks necessary to obtain reliable CD spectra become accessible.”*

5. Discussion of chiral properties and measurement of a confined exciton size is nice, indeed! Here I would suggest to introduce a couple of clarifications.

(i) Definition of ‘exciton size’. Some people define this quantity as a spatial distribution of center of mass of exciton (i.e. exciton wavefunction delocalization on the tube). Others think about average distance between photoexcited electron and hole (aka 1-2nm). Both are correct since placement of 2 particles needs to be characterized by 2 numbers. I believe you apply the former definition.

RESPONSE:

As pointed out by the reviewer, there are two possible definitions of the exciton size: either as the spatial extent of the coherently delocalized excitonic state or as the electron-hole distance/Bohr radius (compare *Phys. Rev. B* **2009**, *80*, 081410, *Phys. Rev. Lett.* **2011**, *107*, 127401, *J. Mater. Chem. C* **2013**, *1*, 6499). While the exact size of the E₁₁ exciton is a subject of ongoing debate, we have referenced values from theoretical works that investigated the difference in size between free and confined excitons. However, our conclusions on the change in relative size of the free and confined exciton remain unaffected by the applied definition.

To clarify this aspect, we now unambiguously refer to the *exciton coherence length* where a comparison to theoretical work on exciton size is drawn (page 21 of the revised manuscript).

(ii) Another point that perhaps needs to be clarified is logic behind “CDnorm values for the free and defect localized exciton indicates a contraction of the exciton size by a factor of ~2”, which is rationalized by “the resulting increase in oscillator strength” and finally “the trapped E₁₁* exciton has a reduced, but non-vanishing interaction with the chiral carbon nanotube lattice” To me (as evident from any quantum-chemical calculations), the oscillator strength of E₁₁ exciton is about 2 times larger than that of E₁₁* (i.e. the oscillator strength is proportional to the exciton size). This is barring any momentum considerations and applies only to ‘still’ excitons. I have problems with applying Miyauchi et al. arguments since they are operating with E₁₁ (X₁₁) size of 2 nm and E₁₁* (X₁₁*) size of 0.8 nm. This assumes true quantum confinement (electron and hole are squeezed like in quantum dot) and true 0-dimension system. Altogether this makes appearance of a giant oscillator strength, i.e. an increase of this quantity. Your sizes are different (5-10 vs 2-5 nm), and the electron and hole are not quantum confined. I personally would interpret your results as decrease of CDnorm values by a factor of 2 assumes reduction of the oscillator strength and subsequently a reduced exciton size by factor of 2, while the trapped E₁₁* exciton has a similar to E₁₁ interaction with the chiral carbon nanotube lattice. Again I would look at the orbital picture of E₁₁ and E₁₁* - they have the same wavefunction structure mimicking the tube chirality. I do not think that one needs to be definitive at this point and rather postulate a challenge to subsequent theoretical studies that will follow your work and help rationalize findings.

RESPONSE:

We agree with the reviewer that this aspect requires a more detailed theoretical treatment and **have added a supplementary note (see page 33 of the revised Supplementary Information) and rephrased our discussion on the estimation of free and confined exciton sizes in the main manuscript accordingly.** In the respective supplementary note, an estimate of absorption dissymmetry factors within a semi-quantitative approach is provided, based on the interplay of the rotational and dipole strengths of the E_{11} and E_{11}^* transitions (see also response to comment #3 by reviewer 3).

So far, the current theoretical work on CD in carbon nanotubes could reproduce the chiroptical characteristics of E_{ii}/E_{ij} transitions in pristine SWCNTs (*Nat. Commun.* **2016**, *7*, 12899, *Phys. Rev. B* **2017**, *95*, 155436). **We highlight that our work provides a starting point for further theoretical studies on the CD of functionalized SWCNTs (see page 23 of the revised manuscript).**

Reviewer #2 (Remarks to the Author):

The manuscript by Sebastien et al. demonstrated a novel photocatalytic functionalization method for generating oxygen defects in surfactant functionalized SWCNTs using ROS, anthraquinones, which react with the nanotube lattice under mild conditions. The level of reactivity of the ROS can be controlled easily, offering flexibility and scalability. They show that the approach is suitable for (6,5) purified and non-purified SWCNTs, for other different purified chiralities, enantiomers and even in organic solvents. The paper is well written and demonstrates several methods as a control to prove that oxygen defects occur. Moreover, the approach can be scaled to high SWCNT concentrations and allows spatially resolved functionalization in thin-film SWCNT networks, with potential applications in quantum emitters, bioimaging, and light-emitting devices. Overall, the work presents a scalable, tunable, and biocompatible photocatalytic approach for introducing luminescent oxygen defects in SWCNTs, demonstrating that these defects retain the chiroptical properties of the nanotube. Still, some issues need to be addressed before publication, as detailed below:

1) Details about the excitation\irradiation parameters are missing in the results shown in Fig. 1a (excitation wavelength) and Fig. 1b (duration of UV irradiation).

RESPONSE:

We have added this information in the caption of Fig. 1.

2) The authors mention six-fold enhanced emission of functionalized SWCNT dispersions at identical concentrations. Is this a general result? Which condition was tested? The details are missing in the text. How was this enhancement quantified?

RESPONSE:

The six-fold brightness enhancement was determined from absolute PLQY measurements of pristine and functionalized (6,5) SWCNTs (pristine PLQY ~0.6%, functionalized up to 3.6%, for experimental details; see section *PL quantum yield determination* in the Supplementary Information). This result is valid for aqueous dispersions of ATPE-sorted (6,5) SWCNT species with average lengths of ~600 nm (see response to comment #7 by reviewer 4 and see **edited**

Supplementary Figure 19 for AFM length statistics of pristine SWCNTs), which is a common choice for studies of SWCNT photoluminescence and functionalization. Previously, we have demonstrated that the length of the starting material has a strong influence on the brightening factor (Berger *et al.*, *ACS Nano* **2019**, *13*, 9259), with short SWCNT experiencing a larger increase in brightness upon functionalization, but lower absolute final PLQYs. Thus, for other species and different sorting approaches, absolute PLQYs and brightening factors can be expected to be different.

We have added the information that the brightening factor was determined from PLQY measurements in the respective section (see page 9) of the revised manuscript.

3) In Fig. 1d, a legend with the corresponding irradiation time is missing.

RESPONSE:

Fig. 1d (Raman spectra of pristine and functionalized SWCNTs) now includes a corresponding legend indicating the irradiation time.

4) What is PSB in Fig. S1a?

RESPONSE:

PSB is used as an abbreviation for the *phonon side band* in optical spectra of SWCNTs.

We have added this information to the caption of Supplementary Fig. 1.

5) In Fig. 2a, the excitation wavelength used for each chirality is missing. Moreover, it would be beneficial to show excitation-emission maps of these separated samples before and after functionalization in the SI.

RESPONSE:

All sorted SWCNT species were excited at their respective E_{22} optical transition as determined from UV-Vis-nIR absorption spectra (see Supplementary Fig. 11). **We have added this**

information to the Figure 2a caption, and specified the excitation wavelengths for each SWCNT species in Supplementary Fig. 11. The PLE maps of pristine and functionalized SWCNT species were added as new Supplementary Figures 12 and 13 (see below).

New Supplementary Fig. 12 | PLE maps of different pristine SWCNT species. Aqueous dispersions (0.33% w/v SDS) of (8,3) (a), (7,3) (b), (9,4) (c), (9,2) (d), (10,0) (e), and (10,3) (f) SWCNTs.

New Supplementary Fig. 13 | PLE maps of different functionalized SWCNT species. Aqueous dispersions (0.2% w/v DOC) of (8,3) (a), (7,3) (b), (9,4) (c), (9,2) (d), (10,0) (e), and (10,3) (f) SWCNTs.

6) The authors mention no reaction occurred in toluene. Can the author include the SWCNTs emission spectra before and after UV-irradiation?

RESPONSE:

PL spectra of pristine and treated SWCNTs in toluene dispersions were added as new Supplementary Fig. 22 (see below).

New Supplementary Fig. 22 | Treatment of toluene dispersions of (6,5) SWCNTs with *t*-Bu-AQ. PL spectra of (6,5) SWCNTs (wrapped by PFO-BPy in toluene) before and after treatment with *t*-Bu-AQ and UV-light (365 nm) for 1 h.

7) Additionally, why did the authors use SWCNTs in toluene for the thin-film study, rather than THF?

RESPONSE:

The initial shear force mixing process to obtain monochiral (6,5) SWCNTs is carried out in toluene (see section *Preparation of (6,5) SWCNT dispersions* in the Supplementary Information, and Graf *et al.*, *Carbon* **2016**, *105*, 593). An additional solvent transfer was not required before thin film preparation. After the network is deposited all solvent is removed. The aqueous reaction medium (with AQS) is then placed on top of the network.

8) The authors exchange the functionalization of SWCNTs through dialysis to PEG and DNA. They should include the before- and after-absorption spectra.

RESPONSE:

Corresponding UV-Vis-nIR absorption spectra are now included as edited Supplementary Figures 15b (transfer to PL-PEG₅₀₀₀) and 16b (transfer to (GT)₆-ssDNA). A clear red-shift is evident in the E₁₁ absorption indicating the change in surface coverage.

Edited Supplementary Fig. 15b showing UV-Vis-nIR absorption spectra of (6,5) SWCNTs before and after transfer to PL-PEG₅₀₀₀.

Edited Supplementary Fig. 16b showing UV-Vis-nIR absorption spectra of (6,5) SWCNTs before and after transfer to (GT)₆-ssDNA.

9) The authors mention “After functionalization, the AQS could be easily and completely removed from the dispersions, ensuring biocompatibility of the final sample.” How was this achieved and verified?

RESPONSE:

The successful removal of AQS was confirmed by UV-Vis absorption spectroscopy. No absorption bands of AQS were detectable after two consecutive steps of spin-filtration (compare Supplementary Fig. 5b).

We have added this information after the respective statement (page 13) in the revised manuscript.

Minor points:

- “The corresponding mirror image of a (n,m) nanotube is a (n+m,-m) nanotube, as illustrated for the enantiomers (6,5) and (11,- 5)”. I am assuming the authors intended to refer the reader to a particular figure?

RESPONSE:

This statement was only intended to provide an example for the nomenclature of two “mirror image” SWCNT enantiomers (n,m) and (n+m,-m). **We have rephrased the sentence to avoid the ambiguity that “illustrated” could also refer the reader to a figure.**

- Typo: SWNT (on page 18).
- Typo: investigated It (page 4).
- Typo: R2 (subscript, page 12 in SI)
- In the SI, you refer to the panels as sometimes as a, b, c, and sometimes as (a), (b), (c) in the captions.

RESPONSE:

We thank the reviewer for pointing out these errors and have corrected them.

Reviewer #3 (Remarks to the Author):

This manuscript presents a well-executed and comprehensive study on photocatalytic functionalization of SWCNTs with luminescent oxygen defects, demonstrating for the first time CD associated with these defect states. The methodology is versatile, biocompatible, and broadly applicable across various SWCNT chiralities and environments. The observation that E11* transitions retain the chiroptical properties of the host nanotube is significant and opens new directions for chiral optoelectronics and quantum emitters. However, the manuscript requires revision to clarify several important points before it can be considered for publication.

1) First, the authors should address whether circular dichroism in PL was observed or attempted. While the CD of defect-state absorption is compelling, circular dichroism in PL would provide further evidence of chirality retention in the emissive state and is highly relevant for future applications. If not observed, a brief discussion of experimental limits or constraints would be valuable.

RESPONSE:

We agree that the observation of circularly polarized photoluminescence (CPL) from emissive defect states in functionalized SWCNTs would present a major advancement towards future applications. However, in contrast to CD measurements, CPL measurements of near-infrared molecular emitters with low g_{lum} values are significantly more challenging to perform reliably due to the contribution of multiple possible artifacts (*Nat. Commun.* **2023**, *14*, 1065, *Adv. Mater.* **2023**, *35*, 2302279). The correction of such artifacts requires advanced customized setups designed for high-sensitivity, artifact-free CPL measurements. To date, near-infrared CPL of molecular origin (excluding chiral metamaterials) beyond 1000 nm has been observed mainly in metal complexes (in particular, lanthanide complexes), which provide high CPL brightnesses B_{CPL} ($B_{CPL} = \epsilon_{\lambda} \cdot PLQY \cdot \frac{1}{2} g_{lum}$, compare *Chem. Eur. J.* **2020**, *27*, 2920, *Mater. Today* **2024**, *75*, 309).

To the best of our knowledge, even CPL from pristine SWCNT enantiomers has not been demonstrated yet.

Assuming that for the same optical transition, emission dissymmetry factors typically do not exceed absorption dissymmetry factors (*ChemPhotoChem* **2018**, *2*, 386), we expect a maximum CPL

brightness of $\sim 0.03 \text{ M}^{-1}\cdot\text{cm}^{-1}$ for the E_{11}^* emission of functionalized (6,5) SWCNT enantiomers in aqueous dispersions (based on $\epsilon_{\lambda(E_{11})} = 6700 \text{ M}^{-1}\cdot\text{cm}^{-1}$, see *Nano Lett.* **2014**, *14*, 1530, E_{11}^* PLQYs of $\sim 2\%$, and $g_{\text{lum,theo.}}(E_{11}^*) \leq g_{\text{abs}}(E_{11}^*)$). For comparison, state-of-the-art setups can determine near-infrared CPL for emitters with $B_{\text{CPL}} \geq 0.5 \text{ M}^{-1}\cdot\text{cm}^{-1}$ (*Angew. Chem. Int. Ed.* **2023**, *62*, e202302358), which is more than one order of magnitude higher. Hence, the observation of CPL from pristine or functionalized SWCNTs will remain a steep challenge and has not been attempted here.

2) Second, the authors should explicitly discuss the role of symmetry breaking in enabling CD from localized states. Since chirality in CD arises from broken mirror symmetry, the defect-localized exciton must maintain coupling to the chiral lattice.

RESPONSE:

Clearly, symmetry breaking is a prerequisite for the observation of CD of optical transitions, which, in this case, indicates that the localized exciton still strongly couples to the chiral SWCNT lattice (i.e., whose symmetry is already broken due to the roll-up vector).

We have highlighted the role of symmetry breaking in the retention of chiral information in the defect-localized excitonic state (page 21 of the revised manuscript):

“... Thus, the broken mirror symmetry of the helically chiral SWCNT lattice is maintained even in the confined excitonic states of the quantum defects.”

3) Third, the interpretation of reduced CD_{norm} as a measure of exciton size should be supported by reference to a quantitative or semi-quantitative model. A basic discussion of rotational strength or dipole coupling theory would lend physical credibility to this conclusion.

RESPONSE:

We agree with the reviewer that an expanded theoretical treatment of the relation between normalized CD and exciton size is required.

We have included a semi-quantitative description of the interplay of rotational and dipole strength for the E_{11} and E_{11}^* excitons as a supplementary note, see page 33 of the revised Supplementary Information.

4) Lastly, the authors should consider the possibility that conformational flexibility of functional groups (e.g., rotation around single bonds) may reduce the observed CD. If different functional groups were tested or compared, any trends in CD response should be mentioned. Otherwise, this limitation should be acknowledged as a factor in interpreting the results. In summary, the study is of high quality and interest, but requires clarification and modest theoretical framing to fully support its conclusions. I recommend major revision.

RESPONSE:

While a systematic study of the influence of different functional groups on the degree of E_{11}^* CD is certainly of interest, no other currently available SWCNT functionalization method enables the introduction of sufficiently high levels of luminescent defects in highly concentrated nanotube dispersions with adequate selectivity for a single defect binding configuration. All of these requirements must be fulfilled to obtain sufficiently high E_{11}^* absorption peaks to perform reliable near-infrared CD measurements. Thus, unfortunately, no other functional groups could be tested with regard to their influence on the defect CD. However, as the electron and hole density of the defect-trapped exciton are localized on the nanotube lattice and not on the functional group, we expect only a very limited influence of the type of functional group or its conformational flexibility on the extent of the CD. Notably, in the case of oxygen defects, this conformational flexibility can be ruled out completely due to the nature of the ether-D oxygen defect binding configuration.

To illustrate that in the case of ether-D-type oxygen defects, no bond rotation or conformational flexibility is possible, we have inserted a detailed view of the oxygen defect in Fig. 1a (see below).

Fig. 1a with a detailed illustration of the ether-D oxygen defect (E_{11}^*) binding configuration.

Reviewer #4 (Remarks to the Author):

Comments to the Author(s)

1. In Fig. 1, only in Fig. 1c the concentration of ACQ is given, but in the other figures it is unclear whether it is 6 μM as stated in the text. As the concentration of ACQ in the photocatalyst is a particularly important element of this paper, it is necessary to clearly indicate at which concentration the data in each figure was measured.

RESPONSE:

We have added this information to the individual Figures and to the Figure caption.

2. In page 8, Although the E_{11^*}/E_{11} PL intensity ratio is described as increasing in Fig S3, it is not clear whether the E_{11^*}/E_{11} PL intensity ratio was increasing as there were only separate data for E_{11^*} PL intensity and E_{11} PL intensity. Please add information on the E_{11^*}/E_{11} PL intensity ratio.

RESPONSE: We have added the E_{11^*}/E_{11} PL intensity ratio to Supplementary Fig. S3b (see below).

Supplementary Fig. 3b with additional E_{11^*}/E_{11} PL intensity ratio (lower plot).

3. As shown in Figure S7, an increase in oxygen defects in SWCNTs should increase the fluorescence lifetime, so the value of the E11*/E11 PL intensity ratio, an indicator of oxygen defects, should be given. In addition, the relationship between the fluorescence lifetime, τ and the E11*/E11 PL intensity ratio should be shown. Does a higher E11*/E11 PL intensity ratio, for example as shown in Fig 1a, result in a longer fluorescence lifetime?

RESPONSE:

One has to distinguish the PL lifetimes of the intrinsic E11 emission (at ~990 nm) and the PL lifetime of the E11* defect emission (at ~1120 nm). As the PL lifetime of the E11 emission of both pristine and functionalized SWCNTs in dispersion is on the order of 10 ps (*Phys. Rev. Lett.* **2005**, *95*, 247402) it cannot be measured with our TCSPC setup (shorter than instrument response function). For the much longer PL lifetime of the oxygen defects (detection of photons at ~1120 nm), we can obtain values of 150 – 160 ps (as illustrated in Supplementary Fig. 7b). It was shown previously that the PL lifetime of the defect emission depends weakly on the degree of functionalization (*ACS Nano* **2016**, *10*, 8355) and actually decreases at higher levels of nanotube functionalization. Thus, an increase in oxygen defects will lead to higher E11*/E11 PL intensity ratios, but not to longer E11* PL lifetimes.

4. In page 12, the surface coverage of SWCNTs by PL-PEG or ssDNA is determined solely by the red-shift in the fluorescence spectrum, but additional information on the uniformity and thickness of those coatings should be added by morphological analysis, such as TEM or SEM. This is because the uniformity and thickness of the coating is expected to affect optical properties such as E11*/E11 PL intensity ratio.

RESPONSE:

We agree that an analysis of the morphology of wrapping polymers and surfactant shells around carbon nanotubes would be highly interesting. However, standard AFM, TEM and SEM are not suitable to identify the (possibly non-periodic) wrapping geometry of PL-PEG₅₀₀₀ or (GT)₆-ssDNA around (6,5) SWCNTs (diameter < 1 nm) especially under the conditions of the reaction, meaning in water. Generally, the investigation of nanotube coverage by surfactants and changes in hydration is a challenging field of research on its own (see e.g. *Nat. Commun.* **2024**, *15*, 6770) and not the

aim of this work. The only reports that successfully identified the wrapping geometry on SWCNT surfaces, in the form of periodic ssDNA wrapping, were only recently conducted using cryo-EM and special high-resolution AFM (*Science* **2022**, *377*, 535, *Sci. Adv.* **2025**, *11*, eadt9844, respectively), which are sophisticated techniques that are not available to us. We doubt that these kinds of analyses could provide meaningful insights with regard to the reactivity of PL-PEG₅₀₀₀- or (GT)₆-wrapped SWCNTs towards reactive oxygen species in water. Furthermore, these measurements only provide the wrapping geometry in the solid phase, whereas the functionalization reaction occurs in the liquid phase.

For the purpose of demonstrating the general feasibility of functionalization of PL-PEG₅₀₀₀- and ssDNA-wrapped (6,5) SWCNTs with reactive oxygen species photocatalytically produced by anthraquinones, we believe that it is sufficient to show their successful application to SWCNT dispersions with these wrapping polymers by means of PL and absorption spectroscopy (see **new Supplementary Figs. 15b and 16b**) The spectral shift of the E₁₁ transition is the currently accepted method for confirming surfactant exchange in SWCNT dispersions (see e.g. *Nat. Commun.* **2016**, *7*, 10241, *J. Phys. Chem. C* **2020**, *124*, 9045, *Anal. Chem.* **2021**, *93*, 6446, *J. Phys. Chem. C* **2024**, *128*, 13064).

The influence of different surfactants on the accessibility of the nanotube lattice for reactive species has been demonstrated previously, and it is clear that a difference in coverage will significantly influence the E₁₁*/E₁₁ PL intensity ratio (*Nat. Commun.* **2019**, *10*, 2874, *ACS Nano* **2024**, *18*, 20667).

We have expanded our statement on this possible contribution to the differences in observed PL spectra in the main text (page 13 of the revised manuscript).

“...The apparently lower reactivity of ssDNA-wrapped SWCNTs may result from a difference in nanotube surface coverage⁵⁸. Possible unwanted side reactions between ROS and the buffer system (PBS) may also contribute in addition to a lower colloidal stability due to alterations of DNA strands by ROS. The functionalization efficiency might be improved by optimizing the oligomer sequence or ssDNA concentration⁵⁹.”

5. In addition to the above, it is essential to explain the clear reason for the difference in E_{11}^*/E_{11} PL intensity ratio between PEG-coated SWCNTs and DNA-coated SWCNTs.

RESPONSE:

Both PL-PEG₅₀₀₀ and ssDNA are expected to have a higher surface coverage of SWCNTs than SDS (*Nat. Commun.* **2016**, *7*, 10241), which will generally lead to a decrease in reactivity compared to our reference system (i.e. to (6,5) SWCNTs in ~0.3% SDS). However, we expect that other factors will also contribute to the apparent reactivity of the system. First, the functionalization of PL-PEG₅₀₀₀-stabilized SWCNTs was performed in deionized, purified water, whereas dispersions of (GT)₆-ssDNA-wrapped SWCNTs were prepared with 1x PBS buffer solution in order to ensure stability of the ssDNA strands. As the functionalization reaction is based on the creation of reactive oxygen species, their reaction with components of the buffer system could decrease their concentration thus decreasing the number of defects created on the SWCNT sidewall. Additionally, reactions between reactive oxygen species and ssDNA strands (*Biochemistry* **1990**, *29*, 8017, *Proc. Natl. Acad. Sci. U.S.A.* **1998**, *95*, 9738) will decrease the concentration of available radical species, and possibly even reduce the stability of the ssDNA wrapping itself.

We now point out the possibility of side reactions contributing to lower E_{11}^*/E_{11} PL intensity ratios and a decrease in stability of the dispersion in the respective section of the main text (page 13 of the revised manuscript).

6. In Fig. 2c, a legend similar to Fig. 2b should be written to explain each fluorescence spectrum and an explanation should be added in the text.

RESPONSE:

Due to lack of free space in Fig. 2c, we have added information on the irradiation conditions as text in the Figure caption.

7. In Figure S17, AFM images after sonication as well as AFM images before sonication should be appended for comparison. In addition, zoomed-in images of about 1 $\mu\text{m} \times 1 \mu\text{m}$ are also needed to show fragmented SWCNTs of several tens of nm.

RESPONSE:

We have performed additional statistical AFM length analyses for SWCNTs before tip sonication and added the results together with zoomed-in images ($1\ \mu\text{m} \times 1\ \mu\text{m}$) of shortened SWCNTs to Supplementary Fig. 19 (see below).

Edited Supplementary Fig. 19 with atomic force micrograph of pristine (6,5) SWCNTs before tip sonication (a), corresponding length histogram (b), and zoomed-in atomic force micrographs of shortened SWCNTs (e).

8. The E₁₁* signal of the pre-functionalized (6,5) SWCNTs of the PLM maps shown in Fig S23 is already stronger than for the enantiomeric (11,-5) SWCNTs, what is the origin of this difference? Also, although CD and PL of the enantiomeric functionalized SWCNTs in E₁₁* transitions shown in Fig 4c and d have been compared, the reviewer considers that the correct assessment may not be possible due to enantiomeric differences in the E₁₁* signal in the pre-functionalised SWCNTs.

RESPONSE:

PL spectra of the (6,5) SWCNT enantiomers show a small E₁₁* PL peak (E₁₁*/E₁₁ PL intensity ratio of 0.28), which is slightly lower for the (11,-5) enantiomer (E₁₁*/E₁₁ PL intensity ratio 0.16). This difference can be attributed to small deviations of the density of unintentional defects in the nanotube raw material and the possibility of very few luminescent defects (1-2 μm⁻¹) being introduced by sonication treatment and exposure to air. Note that even a few luminescent defects will result in noticeable PL signals due to funneling of mobile excitons to these defects. Compared to the final E₁₁*/E₁₁ PL intensity ratios of ~30:1, the relative difference for the pristine SWCNT enantiomers is negligible.

Importantly, these differences in PL spectra are not reflected in UV-Vis-nIR absorption or CD spectra of the two pristine enantiomers (due to the very low density and lower oscillator strength of defects). The absorption at the E₁₁* optical transition is essentially identical for both nanotube enantiomers prior to functionalization (compare Fig. 4b in the main manuscript and Supplementary Fig. 25). Therefore, the interpretation of CD spectra and analysis of chiroptical properties of luminescent defects remain unaffected by this small deviation in the PL spectra of pristine SWCNTs. **We added a corresponding note to the caption of Supplementary Fig. 26.**

9. In page 10, the fluorescence lifetime values in SI Fig 7 (158 ps) differ from the values (153 ps) in the text. Please correct either to the correct value. In page 20, the figure 4c is mistakenly written as Figure 4d.

RESPONSE:

We thank the reviewer for pointing out these errors and have corrected them.

Additional changes

We realized that in the initially submitted version of the Supplementary Information, the PL intensities in Supplementary Fig. 8c were both normalized to the initial E_{11} PL intensity instead of the E_{11} and E_{11}^* PL intensity, and their labelling was erroneously interchanged. **We have included the corrected Figure in the revised Supplementary Information (see below for a comparison of the previous and updated version of Supplementary Figure 8c).**

c

c

Earlier version of Supplementary Fig. 8c (left) / corrected and updated Supplementary Fig. 8c (right).

Point-by-Point Response
Manuscript # NCOMMS-25-07588A

REVIEWERS' COMMENTS

Reviewer #1 (Remarks to the Author):

The authors have adequately addressed all of the reviewers' comments, and I believe the manuscript is now suitable for publication in Nature Communications in its current form.

Reviewer #2 (Remarks to the Author):

The authors have addressed all my comments, and I am happy to recommend publication.

Reviewer #3 (Remarks to the Author):

As Reviewer #3, I have carefully reviewed the authors' rebuttal and revised manuscript. I am pleased to report that the authors have satisfactorily addressed all my concerns through their detailed responses and manuscript revisions. The additional analyses and clarifications provided have strengthened the manuscript's scientific rigor and clarity. Based on these improvements, I strongly recommend publication of this work in Nature Communications. The manuscript now presents a clear and compelling contribution to the field.

RESPONSE:

We thank all reviewers for their swift and positive assessment of the revised manuscript. The editorial requests have been included as requested in final revised documents

(see Editorial_requests_checklist_NCOMMS-25-07588A_Zaumseil_1746099796_25.docx)